# MISSING MASS FOR DIFFERENTIALLY PRIVATE DOMAIN DISCOVERY

**Travis Dick**
Google Research

**Matthew Joseph**
Google Research

**Vinod Raman** *
Department of Statistics
University of Michigan, Ann Arbor

## ABSTRACT

We study several problems in differentially private domain discovery, where each user holds a subset of items from a shared but unknown domain, and the goal is to output an informative subset of items. For set union, we show that the simple baseline Weighted Gaussian Mechanism (WGM) has a near-optimal $\ell_1$ missing mass guarantee on Zipfian data as well as a distribution-free $\ell_\infty$ missing mass guarantee. We then apply the WGM as a domain-discovery precursor for existing known-domain algorithms for private top-$k$ and $k$-hitting set and obtain new utility guarantees for their unknown domain variants. Finally, experiments demonstrate that all of our WGM-based methods are competitive with or outperform existing baselines for all three problems.

## 1 INTRODUCTION

Modern data analysis often requires working in data domains like queries, reviews, and purchase histories that are *a priori* unknown or impractically large (e.g., the set of all strings up to a fixed length). For these datasets, domain discovery is a critical first step for efficient downstream applications. Differential privacy (Dwork et al., 2006) (DP) enables privacy-preserving analysis of sensitive data, but it complicates domain discovery.

In the basic problem of set union, for example, each user has a set of items, and the goal is simply to output as many of these items as possible. This is a necessary step before further analysis, so set union (also known as key selection or partition selection) is a core component of several industrial (Wilson et al., 2020; Rogers et al., 2021; Amin et al., 2023) and open source (OpenDP, 2025) DP frameworks. For similar reasons, there are by now many DP set union algorithms in the literature. However, there are almost no provable utility guarantees (see Section 1.1). This makes it difficult to understand how well existing algorithms work, or how much they can be improved.

**Our Contributions.** We prove utility guarantees for several problems in DP domain discovery. First, by reframing DP set union in terms of mass instead of cardinality (i.e., the fraction of all items recovered, rather than the number of unique items), we prove utility guarantees for the simple and scalable Weighted Gaussian Mechanism (Gopi et al., 2020) (WGM). We first show that the WGM has near-optimal $\ell_1$ missing mass on Zipfian data (Theorem 3.3). We then prove a similar but distribution-free $\ell_\infty$ missing mass guarantee (Theorem 3.6).

Next, we build on these results by considering unknown domain variants of top-$k$ and $k$-hitting set and obtain further utility guarantees for simple algorithms that run WGM to compute a baseline domain and then run a standard known-domain algorithm afterward. Relying on Theorem 3.6 enables us to prove utility guarantees for both top-$k$ (Theorem 4.3) and $k$-hitting set (Theorem 4.5).

Finally, we evaluate our algorithms against the existing state of the art on six real-world datasets from varied domains (Section 5). These experiments demonstrate that, in addition to their theoretical guarantees, our WGM-based methods obtain strong empirical utility.

---

*Work done while interning at Google Research.

## 1.1 RELATED WORK

**DP Set Union.** Early work by Korolova et al. (2009) introduced the core idea of collecting a bounded number of items per user, constructing a histogram of item counts, and releasing items whose noisy counts exceed a carefully chosen threshold. Desfontaines et al. (2022) developed an optimal algorithm for the restricted setting where each user contributes a single item. Gopi et al. (2020) adapted noisy thresholding in the Weighted Gaussian Mechanism (WGM) by scaling contributions to unit $\ell_2$ norm. Swanberg et al. (2023) investigated a repeated version of WGM, and Chen et al. (2025) further built on these ideas by incorporating adaptive weighting to determine user contributions, and proved that the resulting algorithm dominates the WGM (albeit by a small margin, empirically). A separate line of work has studied sequential algorithms that attempt to choose user contributions adaptively, obtaining better empirical utility at the cost of scalability (Gopi et al., 2020; Carvalho et al., 2022).

We note that, as the utility results of Desfontaines et al. (2022) and Chen et al. (2025) are stated relative to other algorithms, our work is, to the best of our knowledge, the first to prove absolute utility guarantees for DP set union.

**DP Top-$k$.** While several algorithms have been proposed for retrieving a dataset's $k$ most frequent items given a known domain (Bhaskar et al., 2010; McKenna & Sheldon, 2020; Qiao et al., 2021; Gillenwater et al., 2022), to the best of our knowledge only Durfee & Rogers (2019) provide an algorithm for the unknown domain setting (see discussion in Section 5.2). They also provide a utility guarantee in terms of what Gillenwater et al. (2022) call $k$-relative error, which bounds the gap between the smallest-count output item and the $k^{th}$ highest-count item. In contrast, we prove a utility result for the more stringent notion of missing mass (see discussion in Section 4.1).

**DP $k$-Hitting Set.** In $k$-hitting set, the objective is to output a set of $k$ items that maximizes the number of users whose subset intersects with it. This problem can also be viewed as an instance of cardinality-constrained submodular maximization. Previous works on private submodular maximization by Mitrovic et al. (2017) and Chaturvedi et al. (2021) establish approximation guarantees in the known-domain setting. However, they are not directly applicable when the domain is unknown.

## 2 PRELIMINARIES

### 2.1 NOTATION

Let $\mathcal{X}$ denote a countable universe of items. A dataset $W$ of size $n$ is a collection of subsets $\{W_i\}_{i \in [n]}$ where $W_i \subset \mathcal{X}$ and $|W_i| < \infty$. We will use $N = \sum_i |W_i|$ to denote the total number of items across all users in the dataset and $M := |\bigcup_i W_i|$ to denote the number of unique items across $W$. For an element $x \in \bigcup_i W_i$, we let $N(x) := \sum_i \mathbf{1}\{x \in W_i\}$ denote its frequency. For a number $r \in [M]$, we use $N_{(r)}$ to denote the $r$'th largest frequency after sorting $\{N(x)\}_{x \in \bigcup_i W_i}$ in decreasing order. We will use $\mathcal{N}(0, \sigma^2)$ to denote a mean-zero Gaussian distribution with standard deviation $\sigma$. Finally, we use the notation $\tilde{O}_k(\cdot)$ to suppress poly-logarithmic factors in $k$ and likewise for $\tilde{\Omega}_k$ and $\tilde{\Theta}_k$.

### 2.2 DIFFERENTIAL PRIVACY

We say that a pair of datasets $W, W'$ are neighboring if $W'$ is the result of adding or removing a single user from $W$. In this work, we consider randomized algorithms $\mathcal{A} : (2^{\mathcal{X}})^\star \rightarrow 2^{\mathcal{X}}$ which map a dataset $W$ to a random subset $S \subseteq \mathcal{X}$. We say that $\mathcal{A}$ is $(\epsilon, \delta)$-differentially private if its distribution over outputs for two neighboring datasets are "close."

**Definition 2.1** (Dwork et al. (2006)). *A randomized algorithm $\mathcal{A}$ is $(\epsilon, \delta)$-differentially private if for all pairs of neighboring datasets $W, W'$, and all events $Y \subseteq 2^{\mathcal{X}}$,*

$$\mathbb{P}_{S \sim \mathcal{A}(W)}[S \in Y] \leq e^\epsilon \mathbb{P}_{S' \sim \mathcal{A}(W')}[S' \in Y] + \delta.$$

We only consider approximate differential privacy ($\delta > 0$) since we will often require that $\mathcal{A}(W) \subseteq \bigcup_i W_i$, which precludes pure differential privacy ($\delta = 0$).

## 2.3 Private Domain Discovery and Missing Mass

In private domain discovery, we are given a dataset $W$ of $n$ users, each of which holds a subset of items $W_i \subseteq \mathcal{X}$ such that $|W_i| < \infty$ and $\mathcal{X}$ is unknown. Given $W$, our is goal is to extract an *informative* subset $S \subseteq \bigcup_i W_i$ that captures the "domain" of $W$ while preserving differential privacy. In this paper, we often measure quality in terms of the *missing mass* of $S$.

**Definition 2.2.** *Given dataset $W$ and output set $S$, the* missing mass *of $S$ with respect to $W$ is*

$$\mathrm{MM}(W, S) := \sum_{x \in \bigcup_i W_i \setminus S} \frac{N(x)}{N}.$$

Smaller values of $\mathrm{MM}(W, S)$ indicate the that $S$ better captures the high-frequency items in $\bigcup_i W_i$. A useful perspective to the MM is that it is the $\ell_1$ norm of the vector $(N(x)/N)_{x \in \bigcup_i W_i \setminus S}$. This view yields a generalization of the MM objective by taking the $p$'th norm of the vector of missing frequencies. That is, for $p \geq 0$, define

$$\mathrm{MM}_p(W, S) := \left\| \left( \frac{N(x)}{N} \right)_{x \in \bigcup_i W_i \setminus S} \right\|_p \tag{1}$$

where $\| \cdot \|_p : \mathbb{R}^\star \to \mathbb{R}$ denotes the $\ell_p$ norm. The usual missing mass objective corresponds to setting $p = 1$. However, it is also meaningful to set $p \neq 1$. For example, when $p = \infty$, the objective corresponds to minimizing the maximum missing mass. When $p = 0$, we recover the cardinality-based objective studied by existing work (see Related Work).

## 3 Private Set Union

In general, we consider algorithms that satisfy the following "soundness" property: an item only appears in the output if it also appears in the input dataset.

**Assumption 1.** *For every algorithm $\mathcal{A} : (2^{\mathcal{X}})^\star \to 2^{\mathcal{X}}$ and dataset $W$, we require $\mathcal{A}(W) \subseteq \bigcup_i W_i$.*

Assumption 1 is standard across works in the unknown domain setting. However, even with this assumption, it is difficult to obtain a meaningful trade-off between privacy and missing mass without assumptions on $W$.

To see why, fix some $n \in \mathbb{N}$, and consider the singleton dataset $W$ where each user has a single, unique item, such that $W_i = \{x_i\}$ and $x_i \neq x_j$ for $i \neq j$. Fix $j \in [n]$ and consider the neighboring dataset $W'$ obtained by removing $W_j$ from $W$. Since we require that $\mathcal{A}(W') \subseteq \bigcup_i W_i'$, we have that $\mathbb{P}\left[x_j \in \mathcal{A}(W')\right] = 0$. Since $\mathcal{A}$ is $(\epsilon, \delta)$-differentially private, we know that $\mathbb{P}\left[x_j \in \mathcal{A}(W)\right] \leq \delta$. Since $j \in [n]$ was picked arbitrarily, we know that this is true for all $j \in [n]$ and hence, $\mathbb{E}_{S \sim \mathcal{A}(W)}\left[\mathrm{MM}(\mathcal{A}, S)\right] \geq 1 - \delta$. As $\delta$ is usually picked to be $o\left(\frac{1}{n}\right)$, it is not possible to significantly minimize MM for these datasets.

Fortunately, in practice, these sorts of pathological datasets are rare. Instead, datasets often exhibit what is known as *Zipf's* or *Power* law (Zipf, 1949; Gabaix, 1999; Adamic & Huberman, 2002; Piantadosi, 2014). This means that the frequency of items in a dataset exhibit a polynomial decay. Hence, one natural way of measuring the complexity of $W$ is by how "Zipfian" it is.

**Definition 3.1** ($(C, s)$-Zipfian)**.** *Let $C \geq 1$ and $s \geq 0$. A dataset $W$ is $(C, s)$-Zipfian if $\frac{N_{(r)}}{N} \leq \frac{C}{r^s}$ for all $r \in [M]$, where $N_{(r)}$ is the $r$'th largest frequency and $N$ is the total number of items in $W$.*

In light of the hardness above, we first restrict our attention to datasets that are $(C, s)$-Zipfian for $s > 1$. When $s \leq 1$, the hard dataset previously outlined becomes a valid Zipfian dataset. We note that this restriction only impacts how we define the utility guarantee of the algorithm, and not its privacy guarantee; differential privacy is still measured with respect to the *worst-case* pair of neighboring datasets.

As $s$ increases, the empirical mass gets concentrated more and more at the highest frequency item. Accordingly, any upper bound on missing mass should ideally decay as $s$ increases. Another important property of $(C, s)$-Zipfian datasets $W$ is that they restrict the size of any individual set $W_i$. Lemma 3.1, whose proof is in Appendix C.1, makes this precise.

**Lemma 3.1.** *Let $W$ be any $(C, s)$-Zipfian dataset . Then, $\max_i |W_i| \leq (CN)^{1/s}$.*

The rest of this section uses these two properties of Zipfian-datasets to obtain high-probability upper bounds on the missing mass. Our main focus will be on a simple mechanism used in practice known as the Weighted Gaussian Mechanism (WGM) (Gopi et al., 2020).

## 3.1 The Weighted Gaussian Mechanism

The WGM is parameterized by a noise-level $\sigma > 0$, threshold $T \geq 1$, and user contribution bound $\Delta_0 \geq 1$. Given a dataset $W$, the WGM operates in three stages. In the first stage, the WGM constructs a random dataset by subsampling without replacement from each user's itemset to ensure that each user has at most $\Delta_0$ items. In the second, stage the WGM constructs a weighted histogram over the items in the random dataset. In the third stage, the WGM computes a noisy weighted histogram by adding mean-zero Gaussian noise with standard deviation $\sigma$ to each weighted count. Finally, the WGM returns those items whose noisy weighted counts are above the threshold $T$. Pseudocode appears in Algorithm 1.

---

**Algorithm 1** Weighted Gaussian Mechanism

---

**Input:** Dataset $W$, noise level $\sigma$, threshold $T$, and user contribution bound $\Delta_0$.

1 Construct random dataset $\widetilde{W}$ such that for every $i \in [n]$, $\widetilde{W}_i \subseteq W_i$ is a random sample (without replacement) of size $\min\{\Delta_0, |W_i|\}$ from $W_i$.

2 Compute weighted histogram $\widetilde{H} : \bigcup_i \widetilde{W}_i \to \mathbb{R}$ such that, for each $x \in \bigcup_i \widetilde{W}_i$,

$$\widetilde{H}(x) = \sum_{i=1}^n \left( \frac{1}{|\widetilde{W}_i|} \right)^{1/2} \mathbf{1}\{x \in \widetilde{W}_i\}.$$

3 For each $x \in \bigcup_i \widetilde{W}_i$, sample $Z_x \sim \mathcal{N}(0, \sigma^2)$ and compute noisy $\widetilde{H}'(x) := \widetilde{H}(x) + Z_x$.

4 Keep items with large noisy weighted counts $S = \left\{ x \in \bigcup_i \widetilde{W}_i : \widetilde{H}'(x) \geq T \right\}$.

**Output:** $S$

---

The following theorem from Gopi et al. (2020) verifies the approximate DP guarantee for the WGM.

**Theorem 3.2** (Theorem 5.1 (Gopi et al., 2020)). *For every $\Delta_0 \geq 1$, $\epsilon > 0$ and $\delta \in (0, 1)$, if $\sigma, T > 0$ are chosen such that*

$$\Phi\left( \frac{1}{2\sigma} - \epsilon\sigma \right) - e^\epsilon \Phi\left( -\frac{1}{2\sigma} - \epsilon\sigma \right) \leq \frac{\delta}{2} \quad \text{and} \quad T \geq \max_{1 \leq t \leq \Delta_0} \left( \frac{1}{\sqrt{t}} + \sigma \Phi^{-1}\left( \left(1 - \frac{\delta}{2}\right)^{\frac{1}{t}} \right) \right)$$

*then the WGM run with $(\sigma, T)$ and input $\Delta_0$ is $(\epsilon, \delta)$-differentially private.*

In Appendix C.2.1, we prove that the smallest choice of $\sigma$ and $T$ to satisfy the constraints in Theorem 3.2 gives that $\sigma = \Theta\left( \frac{1}{\epsilon} \sqrt{\log(1/\delta)} \right)$ and $T = \Theta\left( \max\left\{ \sigma\sqrt{\log\left(\frac{\Delta_0}{\delta}\right)}, 1 \right\} \right) = \tilde{\Theta}_{\delta,\Delta_0}(\max\{\sigma, 1\})$. This result will be useful for deriving asymptotic utility guarantees involving the WGM.

## 3.2 Upper bounds on Missing Mass

Our main result in this section is Theorem 3.3, which provides a high-probability upper bound on the missing mass for the WGM in terms of the Zipfian parameters of the input dataset.

**Theorem 3.3.** *For every $s > 1$, $C \geq 1$ and $(C, s)$-Zipfian dataset $W$, if the WGM is run with noise parameter $\sigma > 0$, threshold $T \geq 1$, and user contribution bound $\Delta_0 \geq 1$, then with probability at least $1 - \beta$ over $S \sim \text{WGM}(W, \Delta_0)$, we have that*

$$\text{MM}(W, S) = \tilde{O}_{\beta,C,N} \left( \frac{C^{\frac{1}{s}}}{s - 1} \left( \frac{\max_i |W_i|}{N\sqrt{q^\star}} \right)^{\frac{s-1}{s}} (T + \sigma)^{\frac{s-1}{s}} \right).$$

*where $q^\star := \min\{\max_i |W_i|, \Delta_0\}$.*

Note that in Theorem 3.3 the missing mass decays as the total number of items $N$ grows. Moreover, as $C$ decreases or $s$ increases, the upper bound on missing mass decreases when $N$ is sufficiently large compared to $\sigma$ and $T$. This matches our intuition, as decreasing $C$ and increasing $s$ results in datasets that exhibit faster decays in item frequencies so relatively more of the mass is contained in high-mass items.

The proof of Theorem 3.3 relies on three helper lemmas. Lemma C.2 provides an upper bound on the missing mass due to the subsampling stage. Lemma C.3 guarantees that a high-frequency item in the original dataset will remain high-frequency in the subsampled dataset. Finally, Lemma C.4 provides a high-probability upper bound on the frequency of items that are missed by the WGM during the thresholding step. We provide the full proof in Appendix C.2.2.

Theorem 3.3 bounds the overall missing mass of the WGM mechanism. As corollary, note that if $\Delta_0 \geq \max_i |W_i|$ then the missing mass contributed by the subsampling step vanishes. By Theorem 3.2 and Lemma C.1, for every user contribution bound $\Delta_0 \geq 1$, we need to pick $\sigma = \Theta\left(\frac{1}{\epsilon}\sqrt{\log(1/\delta)}\right)$ and $T = \tilde{\Theta}_{\Delta_0,\delta}(\max\{\sigma, 1\})$ to to achieve $(\epsilon, \delta)$-differential privacy. Substituting these values into Theorem 3.3 gives the following corollary.

**Corollary 3.4.** *In the setting of Theorem 3.3, if we choose the minimum $\sigma$ and $T$ to ensure $(\epsilon, \delta)$-DP, then with probability at least $1 - \beta$, we have that*

$$\mathrm{MM}(W, S) \leq \tilde{O}_{\beta,\delta,\Delta_0,C,N}\left(\frac{C^{\frac{1}{s}}}{s-1}\left(\frac{\max_i |W_i|}{\epsilon N \sqrt{q^\star}}\right)^{\frac{s-1}{s}}\right),$$

*where $q^\star = \min\{\Delta_0, \max_i |W_i|\}$.*

Corollary 3.4 shows that the error due to subsampling can dominate the missing mass. Accordingly, one should aim to set $\Delta_0$ as close as possible to $\max_i |W_i|$. In fact, if one has apriori *public* knowledge of $\max_i |W_i|$, then one should set $\Delta_0 = \max_i |W_i|$. By Lemma 3.1, for any $(C, s)$-Zipfian dataset $W$, $\max_i |W_i| \leq (CN)^{1/s}$ and hence the loss due to setting $\Delta_0$ will only be logarithmic in $N$. However, Corollary 3.4 omits logarithmic factors in $\Delta_0$, so one should avoid $\Delta_0 \gg \max_i |W_i|$.

Theorem 3.5, whose proof is in Appendix D.1, shows that the dependence of $\epsilon$ and $N$ in our upper bound from Corollary 3.4 can be tight.

**Theorem 3.5.** *Let $\mathcal{A}$ be any $(\epsilon, \delta)$-differentially private algorithm satisfying Assumption 1. For every $s > 1, C \geq 1$, there exists a $(C, s)$-Zipfian dataset $W^\star$ such that*

$$\mathbb{E}_{S \sim \mathcal{A}(W^\star)}[\mathrm{MM}(W^\star, S)] = \Omega\left(\frac{C^{1/s}}{s-1}\left(\frac{1}{\epsilon N}\right)^{(s-1)/s}\ln\left(1 + \frac{e^\epsilon - 1}{2\delta}\right)^{(s-1)/s}\right).$$

The proof of Theorem 3.5 exploits Assumption 1 by showing that any private algorithm that satisfies Assumption 1 cannot output low-frequency items with high-probability. We end this section by noting that our proof technique in Theorem 3.3 can also give us bounds on the $\ell_\infty$ missing mass (see Equation 1). Note that unlike Theorem 3.3, Theorem 3.6, whose proof is in Appendix C.2.3, does not require the dataset to be Zipfian.

**Theorem 3.6.** *Let $W$ be any dataset. For every $\epsilon > 0$, $\delta \in (0, 1)$, and user contribution bound $\Delta_0 \geq 1$, picking $\sigma = \Theta\left(\frac{1}{\epsilon}\sqrt{\log(1/\delta)}\right)$ and $T = \tilde{\Theta}_{\Delta_0,\delta}(\max\{\sigma, 1\})$ gives that the WGM is $(\epsilon, \delta)$-differentially private and with probability at least $1 - \beta$ over $S \sim \mathrm{WGM}(W, \Delta_0)$, we have*

$$\mathrm{MM}_\infty(W, S) \leq \tilde{O}_{\Delta_0,\delta,\beta}\left(\frac{\max_i |W_i|}{\epsilon N \sqrt{q^\star}}\right),$$

*where $q^\star = \min\{\Delta_0, \max_i |W_i|\}$.*

Upper bounds on the $\ell_\infty$ norm missing mass will be useful for deriving guarantees for the top-$k$ selection (Section 4.1) and $k$-hitting set (Section 4.2) problems.

## 4 APPLYING THE WEIGHTED GAUSSIAN MECHANISM

This section applies WGM to construct unknown domain algorithms for top-$k$ and $k$-hitting set. For both problems, we spend half of the overall privacy budget running WGM to obtain a domain $D$, and

then spend the other half of the privacy budget running a known-domain private algorithm, using domain $D$, for the problem in question. By basic composition, the overall mechanism satisfies the desired privacy budget. Pseudocode for this approach is given in Algorithm 2.

---

**Algorithm 2** Meta Algorithm

---

**Input:** Dataset $W$, noise-level and threshold $(\sigma, T)$, output size $k$, user contribution bound $\Delta_0 \geq 1$, known-domain mechanism $\mathcal{B}$

5  Let $D \leftarrow \mathrm{WGM}(W, \Delta_0)$ be the output of WGM with noise-level and threshold $(\sigma, T)$ and input $\Delta_0$
6  Let $S \leftarrow \mathcal{B}(W, D, k)$ be the output of $\mathcal{B}$ on input $W$ and domain $D$
**Output:** $S$

---

In the next two subsections, we introduce the top-$k$ selection and $k$-hitting set problems, summarize existing known-domain algorithms, and provide the specification of all algorithmic parameters. An important difference between the results in this section and that of Section 3, is that by using $\mathrm{MM}_\infty$ bounds, we no longer require our dataset to be Zipfian in order to get meaningful guarantees.

### 4.1 PRIVATE TOP-$k$ SELECTION

In the DP top-$k$ selection problem, we are given some $k \in \mathbb{N}$ and our goal is to output, in decreasing order, the $k$ largest frequency items in a dataset $W$. Various loss objective have been considered for this problem, but we focus on missing mass.

**Definition 4.1.** *For a dataset $W$, $k \in \mathbb{N}$, $q \leq k$ and ordered sequence of domain elements $S = (x_1, ..., x_q)$, we denote the* top-$k$ missing mass *by*

$$\mathrm{MM}^k(W, S) = \frac{\sum_{i=1}^{k} N_{(i)} - \sum_{i=1}^{q} N(x_i)}{N}.$$

We let the sequence $S$ have length $q \leq k$ because we will allow our mechanisms to output less than $k$ items, which will be crucial for obtaining differential privacy when the domain $\mathcal{X}$ is unknown. Note that $\mathrm{MM}(W, S) = \mathrm{MM}^k(W, S)$ if one takes $k = |\bigcup_i W_i|$. As before, our objective is to design an approximate DP mechanism $\mathcal{B}$ which outputs a sequence $S \subseteq \bigcup_i W_i$ of size at most $k$ that minimizes $\mathrm{MM}^k(W, S)$ with high probability.

To adapt Algorithm 2 to top-$k$, we need to specify a known-domain private top-$k$ algorithm. We use the peeling exponential mechanism (see Algorithm 3) for its simplicity, efficiency, and tight privacy composition. Its privacy and utility guarantees appear in Lemmas 4.1 and 4.2 respectively.

---

**Algorithm 3** Peeling Exponential Mechanism

---

**Input:** Dataset $W$, domain $D$, noise-level $\lambda$, output size $k \leq |D|$
1  Let $N(x) = \sum_{i=1}^{n} \mathbf{1}\{x \in W_i\}$ for $x \in D$.
2  Let $\tilde{N}(x) = N(x) + Z_x$ for $x \in D$ where $Z_x \sim \mathrm{Gumbel}(\lambda)$.
**Output:** Ordered sequence $(x_1, \ldots, x_k)$ such that $\tilde{N}(x_i) = \tilde{N}_{(i)}$ for all $i \in [k]$.

---

**Lemma 4.1** (Corollary 4.1 (Durfee & Rogers, 2019))**.** *For every $\epsilon > 0$, $\delta \in (0, 1)$ and $k \geq 1$, if $\lambda = \tilde{O}_\delta\left(\frac{\sqrt{k}}{\epsilon}\right)$, then Algorithm 3 is $(\epsilon, \delta)$-differentially private.*

**Lemma 4.2.** *For every dataset $W$, domain $D$, $\lambda \geq 1$ and $k \leq |D|$, if Algorithm 3 is run with noise-level $\lambda$, then with probability $1 - \beta$ over its output $S$, we have that*

$$\frac{1}{N}\left(\sum_{x \in \mathcal{T}_k(W,D)} N(x) - \sum_{x \in S} N(x)\right) \leq O\left(\frac{k\lambda}{N} \log \frac{|D|}{\beta}\right),$$

*where $\mathcal{T}_k(W, D) \subseteq D$ is the true set of top-$k$ most frequent items in $D$.*

We provide the exact $\lambda$ to achieve $(\epsilon, \delta)$-differential privacy for Lemma 4.1 in Lemma B.1. The proof of Lemma 4.2 is standard, relies on Gumbel concentration inequalities, and appears in Appendix

C.3.Similar upper bounds for other performance metrics have also been derived (Bafna & Ullman, 2017). With Lemmas 4.2 and 4.1 in hand, using the same choice of $(\sigma, T)$ as in Theorem 3.2 for WGM yields our main result. The proof of Theorem 4.3 can be found in Appendix C.3.

**Theorem 4.3.** *Fix $\epsilon > 0$, $\delta \in (0,1)$, and user contribution bound $\Delta_0 \geq 1$. For every dataset $W$ and $k \geq 1$, if one picks $\sigma = \Theta\left(\frac{1}{\epsilon}\sqrt{\log(1/\delta)}\right)$, $T = \tilde{\Theta}_{\Delta_0, \delta/2}(\max\{\sigma, 1\})$ from Theorem 3.2, and $\lambda = \tilde{\Theta}_{\delta/2}\left(\frac{\sqrt{k}}{\epsilon}\right)$ from Lemma 4.1, then Algorithm 2, run with Algorithm 3, is $(\epsilon, \delta)$-differentially private and with probability $1 - \beta$, its output $S$ satisfies*

$$\mathrm{MM}^k(W, S) \leq \tilde{O}_{\beta, \delta, \Delta_0}\left(\frac{k}{N}\left(\frac{\max_i |W_i|}{\epsilon\sqrt{q^\star}} + \frac{\sqrt{k}\log(M)}{\epsilon}\right)\right),$$

*where $q^\star := \min\{\Delta_0, \max_i |W_i|\}$.*

We end this section by proving that a linear dependence on $\frac{k}{\epsilon}$ on the top-$k$ missing mass is unavoidable for algorithms satisfying Assumption 1 when $\epsilon \leq 1$.

**Corollary 4.4.** *Let $\epsilon \leq 1$ and $\delta \in (0,1)$. Let $\mathcal{A}$ be any $(\epsilon, \delta)$-differentially private algorithm satisfying Assumption 1. Then, for every $k \geq 1$, there exists a dataset $W$ such that*

$$\mathbb{E}_{S \sim \mathcal{A}(W, k)}\left[\mathrm{MM}^k(W, S)\right] \geq \tilde{\Omega}_\delta\left(\frac{k}{\epsilon N}\right).$$

The proof of Corollary 4.4 is in Appendix D.2 and is largely a consequence of Lemma D.1, which was used to prove the lower bound for set union (Theorem 3.5).

## 4.2 PRIVATE $k$-HITTING SET

In the $k$-hitting set problem, our goal is to output a set $S$ of items of size at most $k$ which intersects as many user subsets as possible, which is useful for data summarization and feature selection (Mitrovic et al., 2017). More precisely, our objective is to design an approximate DP mechanism which maximizes the number of hits $\mathrm{Hits}(W, S) := \sum_{i=1}^n \mathbf{1}\{S \cap W_i \neq \emptyset\}$. Since this problem is also NP-hard without privacy concerns (Karp, 1972), we will measure performance relative to the optimal solution, i.e., show that with high probability, our algorithm output $S$ satisfies

$$\mathrm{Hits}(W, S) \geq \gamma \cdot \mathrm{Opt}(W, k) - \mathrm{err}(\epsilon, \delta, k)$$

where $\mathrm{Opt}(W, k) := \arg\max_{S \subseteq \mathcal{X}, |S| \leq k} \mathrm{Hits}(W, S)$ is the optimal value, $\mathrm{err}(\epsilon, \delta, k)$ is an additive error term that depends on problem specific parameters, and $\gamma \in (0, 1)$ is the approximation factor.

Like our algorithm for top-$k$ selection, our mechanism for the $k$-hitting problem will follow the general structure of Algorithm 2. We will take the known-domain algorithm $\mathcal{B}$ to be the privatized version of the greedy algorithm for submodular maximization, as in Algorithm 1 from Mitrovic et al. (2017). This mechanism repeatedly runs the exponential mechanism (equivalently the Gumbel mechanism) to pick an item that hits a large number of users. After each iteration, we remove all users who contain the item output in the previous round and continue until we either have output $k$ items, run out of items, or run out of users, and return the overall set of items. We call this algorithm the User Peeling Mechanism and its pseudo-code is given in Algorithm 4 in Appendix C.4.

By combining this with the same WGM choice of $(\sigma, T)$ as in Theorem 3.2 for the first step of Algorithm 2, we get the main result of this section.

**Theorem 4.5.** *Fix $\epsilon > 0$ and $\delta \in (0,1)$. For every dataset $W$, $k \geq 1$, and user contribution bound $\Delta_0$, if one picks $\sigma = \Theta\left(\frac{1}{\epsilon}\sqrt{\log(1/\delta)}\right)$, $T = \tilde{\Theta}_{\Delta_0, \delta/2}(\max\{\sigma, 1\})$ from Theorem 3.2, and $\lambda = \tilde{\Theta}_{\delta/2}\left(\frac{1}{\epsilon}\sqrt{k}\right)$ from Lemma 4.1, then Algorithm 2, run with Algorithm 4, is $(\epsilon, \delta)$-differentially private and with probability $1 - \beta$, its output $S$ satisfies*

$$\mathrm{Hits}(W, S) \geq \left(1 - \frac{1}{e}\right)\mathrm{Opt}(W, k) - \tilde{O}_{\beta, \delta, \Delta_0}\left(\frac{k \cdot \max_i |W_i|}{\epsilon\sqrt{q^\star}} + \frac{k^{3/2}}{\epsilon}\log\left(Mk\right)\right),$$

*where $q^\star := \min\{\Delta_0, \max_i |W_i|\}$ and $M = |\bigcup_i W_i|$.*

Theorem 4.5, proved in Appendix C.4, gives that if $k$ is not very large (i.e., $\frac{\ln(Mk)}{\ln(M)} \leq \max_i \sqrt{|W_i|}$), then with high probability, the additive sub-optimality gap is on the order of

$$\tilde{O}_{\Delta_0, \delta, \beta, k} \left( \frac{k^{3/2} \cdot \max_i |W_i| \cdot \log(M)}{\epsilon \sqrt{q^\star}} \right).$$

When $|\mathcal{X}| \gg M$, this provides an improvement over Theorem 1 in Mitrovic et al. (2017) whose guarantee is in terms $\log(|\mathcal{X}|)$ and not $\log(M)$.

As in the lower bound proof for top-$k$ selection, we again rely on the work behind Theorem 3.5 to show that one must lose $\frac{k}{\epsilon}$ from the optimal value by restricting the algorithm $\mathcal{A}$ to output a subset of $\bigcup_i W_i$.

**Corollary 4.6.** *Let $\epsilon \leq 1$, $\delta \in (0, 1)$ and $\mathcal{A}$ be any $(\epsilon, \delta)$-differentially private algorithm satisfying Assumption 1. Then, for every $k \geq 1$, there exists a dataset $W$ such that*

$$\mathbb{E}_{S \sim \mathcal{A}(W,k)} \left[ \text{Hits}(W, S) \right] \geq \text{Opt}(W, k) - \tilde{\Omega}_\delta \left( \frac{k}{\epsilon} \right).$$

## 5 EXPERIMENTS

We empirically evaluate our methods on six real-life datasets spanning diverse settings. Informally, Reddit (Gopi et al., 2020), Amazon Games (Ni et al., 2019), and Movie Reviews (Harper & Konstan, 2015) are "large", while Steam Games (Steam, 2025), Amazon Magazine (Ni et al., 2019), and Amazon Pantry (Ni et al., 2019) are "small" (see Appendix E for details). All experiments use a total privacy budget of $(1, 10^{-5})$-DP; additional experiments using $(0.1, 10^{-5})$-DP appear in Appendix F, but are not significantly qualitatively different. Dataset processing and experiment code can be found in the Supplement.

### 5.1 SET UNION

**Datasets.** We evaluate the WGM and baselines on all six datasets, relegating experiments on the small datasets to the Appendix F.1.1 for space.

**Baselines.** The baselines are the Policy Gaussian mechanism from Gopi et al. (2020) and the Policy Greedy mechanism from Carvalho et al. (2022), as these have obtained the strongest (though least scalable) performance in past work. As suggested in those papers, we set the policy hyperparameter $\alpha = 3$ throughout.

**Results.** Figure 1 plots the average MM across 5 trials, for all three mechanisms as a function of $\ell_0$ bound $\Delta_0 \in \{1, 50, 100, 150, 200, 300\}$. Across datasets, we find that the WGM obtains MM within 5% of that of the policy mechanisms, in spite of their significantly more intensive computation. This contrasts with previous empirical results for cardinality, where sequential methods often output $\approx 2X$ more items (see, e.g., Table 2 in Swanberg et al. (2023)). Plots for the small datasets (Appendix F.1.1) show a similar trend.

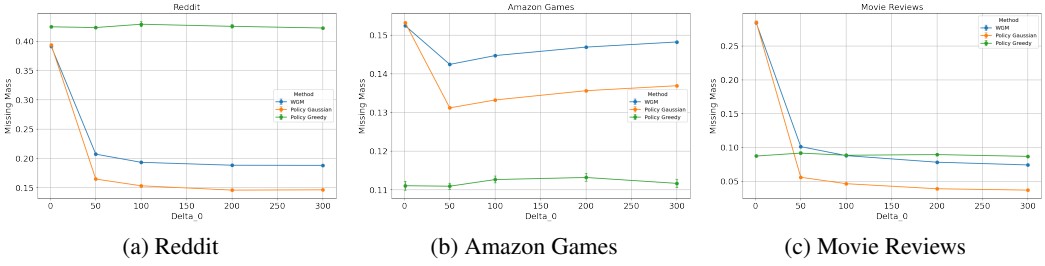

| (a) Reddit | (b) Amazon Games | (c) Movie Reviews |

Figure 1: Set Union MM as a function of $\Delta_0$. Note that lower is better.

## 5.2 Top-$k$

**Datasets.** All methods achieve near $0$ top-$k$ missing mass across all values of number of selected items $k \in \{5, 10, 20, 50, 100, 200\}$ on the three large datasets, as most mass is concentrated in a small number of heavy items. We therefore focus on the three small datasets.

**Baselines.** We compare our WGM-then-top-$k$ mechanism to the limited-domain top-$k$ mechanism from Durfee & Rogers (2019). Unlike our algorithm, the limited-domain mechanism has a hyperparameter $\bar{k}$. As such, for each $k \in \{5, 10, 20, 50, 100, 200\}$, we take as baselines the limited-domain algorithm with $\bar{k} \in \{k, 5k, 10k, \infty\}$. When $\bar{k} < \infty$, we set $\Delta_0 = \infty$ for the limited-domain algorithm. Otherwise, when $\bar{k} = \infty$, we set $\Delta_0 = 100$ for the limited-domain algorithm, as recommended in Section 3 of Durfee & Rogers (2019).

**Results.** Figure 2 compares our method against the limited-domain method across different choices for $k$. Note that each line for the limited-domain method uses a different $\bar{k}$. We find that across all datasets, our method consistently obtains smaller top-$k$ MM than all limited-domain baselines, and its advantage grows with $k$. Plots in Appendix F.2 demonstrate similar trends for a more stringent $\ell_1$ loss.

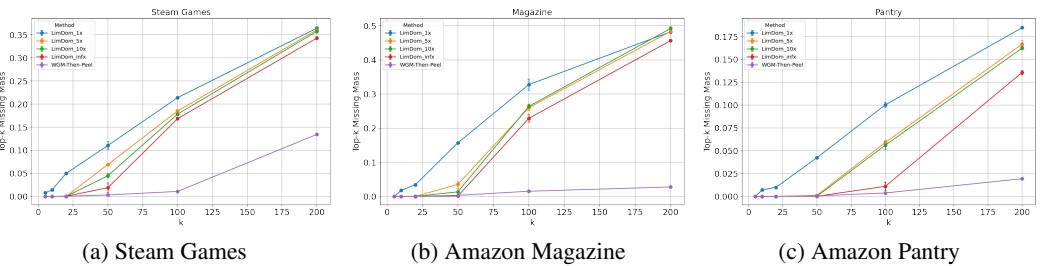

|  (a) Steam Games | (b) Amazon Magazine | (c) Amazon Pantry |

Figure 2: Top-$k$ MM as a function of $k$, using $\Delta_0 = 100$.

## 5.3 $k$-Hitting Set

**Datasets.** We use the same datasets as in the top-$k$ experiments, for the same reason: a small number of items covers nearly all users in the large datasets.

**Baselines.** To the best of our knowledge, there are no existing private algorithm for the $k$-hitting set problem for unknown domains. Hence, we consider the following baselines: the non-private greedy algorithm and the private non-domain algorithm from Mitrovic et al. (2017) after taking $\bigcup_i W_i$ to be a public known-domain. Note that the latter baseline is not a valid private algorithm in the unknown domain setting since, in reality, $\bigcup_i W_i$ is private.

**Results.** Figure 3 plots the average number of users hit, along with its standard error across 5 trials, as a function of $k \in \{5, 10, 20, 50, 100, 200\}$, fixing $\Delta_0 = 100$. We find that our method performs comparably with both baseline methods, neither of which is fully private. In particular, for the Steam Games and Amazon Magazine datasets, our method outperforms the known-domain private greedy algorithm that assumes public knowledge of $\bigcup_i W_i$. This is because our method's application of WGM for domain discovery produces a domain that is smaller than $\bigcup_i W_i$ while still containing high-quality items. This makes an easier problem for the peeling mechanism in the second step.

## 6 Future Directions

We conclude with some possible future research directions. First, our upper and lower bounds for top-$k$ and $k$-hitting set do not match, so closing these gaps is a natural problem. Second, all of our methods enforce $\ell_0$ bounds by uniform subsampling without replacement from each user's item set. Recent work by Chen et al. (2025) employs more involved and data-dependent subsampling strategies to obtain higher cardinality answers. Extending similar techniques to missing mass may be useful.

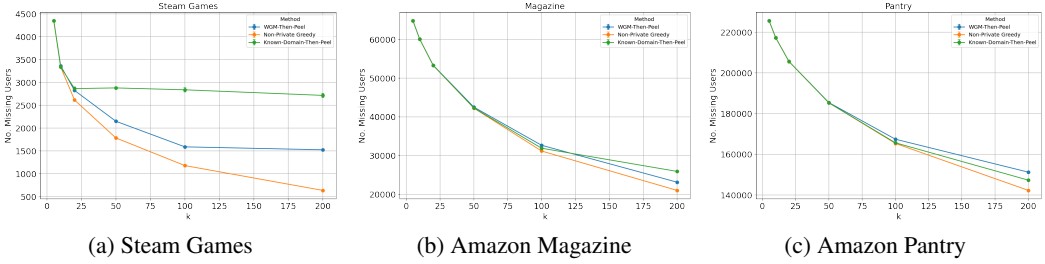

| (a) Steam Games | (b) Amazon Magazine | (c) Amazon Pantry |

Figure 3: Number of missed users as a function of $k$, using $\Delta_0 = 100$.

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

## A  USEFUL CONCENTRATION INEQUALITIES

In this section, we review some basic concentration inequalities that we use in the main text. The first is the following Gaussian concentration equality.

**Lemma A.1** (Gaussian concentration (Vershynin, 2018))**.** *Let $X_1, \ldots, X_n$ be an iid sequence of mean-zero Gaussian random variables with variance $\sigma^2$. Then, for every $\delta \in (0, 1)$, with probability at least $1 - \delta$, we have that*

$$\max_i |X_i| \leq \sigma \sqrt{2 \log \left( \frac{2n}{\delta} \right)}.$$

The second is for the concentration of a sequence of Gumbel random variables. While this result is likely folklore, we provide a proof for completeness.

**Lemma A.2.** *Let $X_1, \ldots, X_n$ be an iid sequence of Gumbel random variables with parameter $\lambda$. Then, for every $\delta \in (0, 1)$, with probability at least $1 - \delta$, we have that*

$$\max_i |X_i| \leq \lambda \cdot \ln \left( \frac{2n}{\delta} \right).$$

*Proof.* Consider a single $X_i$, and recall that the CDF of a Gumbel distribution with parameter $\lambda$ is $F(x) = \exp(-\exp(-x/\lambda))$. Then

$$\begin{aligned}
\mathbb{P}\left[|X_i| > T\right] &= \mathbb{P}\left[X_i > T\right] + \mathbb{P}\left[X_i < -T\right] \\
&= 1 - \exp(-\exp(-T/\lambda)) + \exp(-\exp(T/\lambda)) \\
&\leq \exp(-T/\lambda) + \exp(-\exp(T/\lambda))
\end{aligned}$$

where the inequality uses $1 - e^{-x} \leq x$. Substituting in $T = \lambda \log(2n/\delta)$ yields

$$\mathbb{P}\left[|X_i| > t\right] \leq \frac{\delta}{2n} + \exp(-2n/\delta) \leq \frac{\delta}{n}$$

since for $n \geq 1$ and $\delta \in (0, 1)$, $\frac{2n}{\delta} \geq \ln\left(\frac{2n}{\delta}\right)$. Union bounding over the $n$ samples completes the result. ∎

# B PRIVACY ANALYSIS OF PEELING EXPONENTIAL MECHANISM

**Lemma B.1** (Lemma 4.2 in Gillenwater et al. (2022))**.** *For every* $\epsilon > 0$*,* $\delta \in (0, 1)$ *and* $k \geq 1$*, if* $\lambda = \frac{1}{\epsilon_0}$*, where*

$$\epsilon_0 := \max\left\{\frac{\epsilon}{k}, \sqrt{\frac{8\log\left(\frac{1}{\delta}\right) + 8\epsilon}{k}} - \sqrt{\frac{8\log\left(\frac{1}{\delta}\right)}{k}}\right\},$$

*then Algorithm 3 is* $(\epsilon, \delta)$*-differentially private.*

# C MISSING PROOFS

## C.1 PROOF OF LEMMA 3.1

*Proof.* Let $W$ be a $(C, s)$-Zipfian dataset . Let $r^\star = \max_i |W_i|$. Then, it must be the case that $N_{(r^\star)} \geq 1$. Since $W$ is $(C, s)$-Zipfian, we also know that $N_{(r^\star)} \leq \frac{CN}{(r^\star)^s}$. Hence, we have that $(r^\star)^s \leq CN$ implying that $r^\star \leq (CN)^{1/s}$. ∎

## C.2 PROOFS FOR THE WGM

### C.2.1 PROOF OF $\sigma$ AND $T$

**Lemma C.1.** *For every* $\epsilon > 0$*,* $\delta \in (0, 1)$*, and* $\Delta_0 \geq 1$*, there exists* $\sigma = \Theta\left(\frac{\sqrt{\log\left(\frac{1}{\delta}\right)}}{\epsilon}\right)$ *and*

$T = \tilde{\Theta}_{\delta, \Delta_0}(\max\{\sigma, 1\})$ *which satisfy the conditions in Theorem 3.2.*

*Proof.* Starting with $\sigma$, it suffices to find the smallest $\sigma$ such that

$$\Phi\left(\frac{1}{2\sigma} - \epsilon\sigma\right) \leq \frac{\delta}{2}.$$

By monotonicity of $\Phi^{-1}(\cdot)$, we have that

$$\Phi\left(\frac{1}{2\sigma} - \epsilon\sigma\right) \leq \frac{\delta}{2} \iff \frac{1}{2\sigma} - \epsilon\sigma \leq \Phi^{-1}\left(\frac{\delta}{2}\right).$$

Hence, it suffices to find the smallest $\sigma$ that satisfies

$$2\epsilon\sigma^2 + 2\Phi^{-1}\left(\frac{\delta}{2}\right)\sigma - 1 \geq 0.$$

Using the quadratic formula we can deduce that we need to take

$$\sigma \geq \frac{-\Phi^{-1}\left(\frac{\delta}{2}\right)}{\epsilon} = \frac{\Phi^{-1}\left(1 - \frac{\delta}{2}\right)}{\epsilon} = \Omega\left(\frac{\sqrt{\log\left(\frac{1}{\delta}\right)}}{\epsilon}\right),$$

where the last inequality follows from the fact that $\Phi^{-1}(p) \leq \sqrt{2 \log\left(\frac{1}{1-p}\right)}$ for $p > \frac{1}{2}$. Now for $T$, we have

$$1 + \sigma \Phi^{-1}\left(\left(1 - \frac{\delta}{2}\right)^{\frac{1}{\Delta_0}}\right) \geq \max_{1 \leq t \leq \Delta_0}\left(\frac{1}{\sqrt{t}} + \sigma \Phi^{-1}\left(\left(1 - \frac{\delta}{2}\right)^{\frac{1}{t}}\right)\right).$$

Hence, it suffices to upper bound

$$1 + \sigma \Phi^{-1}\left(\left(1 - \frac{\delta}{2}\right)^{\frac{1}{\Delta_0}}\right).$$

By Bernoulli's inequality and monotonicity of $\Phi^{-1}(\cdot)$, we have that

$$\Phi^{-1}\left(\left(1 - \frac{\delta}{2}\right)^{\frac{1}{\Delta_0}}\right) \leq \Phi^{-1}\left(1 - \frac{\delta}{2\Delta_0}\right).$$

Since $\delta \leq 1$ and $\Delta_0 \geq 1$, we have that

$$\Phi^{-1}\left(1 - \frac{\delta}{2\Delta_0}\right) \leq \sqrt{2 \log\left(\frac{2\Delta_0}{\delta}\right)}.$$

Hence, it suffices to take

$$T = 1 + \sigma\sqrt{2 \log\left(\frac{2\Delta_0}{\delta}\right)} = \Theta\left(\max\left\{\sigma\sqrt{\log\left(\frac{\Delta_0}{\delta}\right)}, 1\right\}\right),$$

This completes the proof. ∎

### C.2.2 PROOF OF THEOREM 3.3

Before we prove Theorem 3.3, we present three helper lemmas, Lemma C.2, C.3, and C.4, which correspond to three different "good" events. Lemma C.2 provides an upper bound on the missing mass due to the subsampling stage. Lemma C.3 guarantees that a high-frequency item in the original dataset will remain high-frequency in the subsampled dataset. Lemma C.4 provides a high-probability upper bound on the frequency of items that are missed by the WGM during the thresholding step. The proof of Theorem 3.3 will then follow by combining Lemmas C.2, C.3, and C.4.

**Lemma C.2.** *Let $W$ be a $(C, s)$-Zipfian dataset for $C \geq 1$ and $s > 1$. Fix a user contribution bound $\Delta_0 \geq 1$. Let $\widetilde{W}$ be the random dataset such that $\widetilde{W}_i \subseteq W_i$ is a random sample without replacement of size $\min\{\Delta_0, W_i\}$. Then, for every $\beta \in (0, 1)$, with probability $1 - \beta$, we have that*

$$\mathrm{MM}(W, \cup_i \widetilde{W}_i) \leq \frac{C^{1/s}}{s - 1}\left(\frac{1}{p^\star N} \log\left(\frac{(CN)^{1/s}}{\beta}\right)\right)^{\frac{s-1}{s}}$$

*where $p^\star := \min\left(1, \frac{\Delta_0}{\max_i |W_i|}\right)$.*

*Proof.* Let $p_i := \min\left(1, \frac{\Delta_0}{|W_i|}\right)$ and $p^\star := \min_i p_i$. Fix an item $x \in \bigcup_i W_i$. Then

$$\mathbb{P}\left[x \notin \bigcup_i \widetilde{W}_i\right] = \prod_{i:x \in W_i}(1 - p_i) \leq \exp\left(-\sum_{i:x \in W_i} p_i\right) \leq e^{-p^\star N(x)},$$

where $N(x) = \sum_i \mathbf{1}\{x \in W_i\}$. Fix $\beta \in (0, 1)$ and consider the threshold

$$Q := \frac{1}{p^\star} \log\left(\frac{(CN)^{1/s}}{\beta}\right).$$

Note that $Q^\star \geq 1$ by definition of $p^\star$. Since $W$ is $(C, s)$-Zipfian, for any $r \in [M]$ with $N_{(r)} > Q$, it must be the case that $r \leq \left(\frac{CN}{Q}\right)^{1/s}$ by $r^s \leq \frac{CN}{N_{(r)}}$. Hence, there are at most $\left(\frac{CN}{Q}\right)^{1/s}$ "heavy" items whose frequencies are above $Q$. By the union bound, we get that

$$\mathbb{P}\left[\exists x \in \bigcup_i W_i \setminus \widetilde{W}_i \text{ and } N(x) > Q\right] \leq \left(\frac{CN}{Q}\right)^{1/s} e^{-p^\star Q} \leq \beta$$

so with probability at least $1 - \beta$, we have that $N(x) > Q \implies x \in \bigcup_i \widetilde{W}_i$ for all $x \in \bigcup_i W_i$. Under this event, we have that

$$\mathrm{MM}(W, \cup_i \widetilde{W}_i) \leq \frac{1}{N} \sum_{x : N(x) \leq Q} N(x) \leq \frac{1}{N} \sum_{r \geq r_0} N_{(r)}$$

where $r_0 = \max\left\{\left(\frac{CN}{Q}\right)^{1/s}, 1\right\}$. Using the fact that $N_{(r)} \leq CNr^{-s}$ and $s > 1$ (by assumption), we get that

$$\frac{1}{N} \sum_{r \geq r_0} N_{(r)} \leq C \sum_{r \geq r_0} r^{-s} \leq C \int_{r_0 - 1}^{\infty} x^{-s} dx = \frac{C}{s-1}(r_0 - 1)^{1-s} \leq \frac{C}{s-1} r_0^{1-s}.$$

Since $r_0 \geq \left(\frac{CN}{Q}\right)^{1/s}$, we get $\mathrm{MM}(W, \cup_i \widetilde{W}_i) \leq \frac{C^{1/s}}{s-1} \left(\frac{Q}{N}\right)^{\frac{s-1}{s}}$. Using $Q = \frac{1}{p^\star} \log\left(\frac{(CN)^{1/s}}{\beta}\right)$ completes the proof. ∎

**Lemma C.3.** *In the same setting as Lemma C.2, for every $\beta \in (0, 1)$, we have that with probability $1 - \beta$, for every $x \in \bigcup_i W_i$,*

$$N(x) \geq \tau_2 \implies \widetilde{N}(x) \geq \frac{1}{2} p^\star N(x),$$

*where $p^\star := \min\left(1, \frac{\Delta_0}{\max_i |W_i|}\right)$, $\tau_2 := \frac{8}{p^\star} \log\left(\frac{(CN)^{1/s}}{\beta}\right)$, and $\widetilde{N}(x) = \sum_i \mathbf{1}\{x \in \widetilde{W}_i\}$.*

*Proof.* Fix some $x \in \bigcup_i W_i$ such that $N(x) \geq \tau_2$. Then, $\widetilde{N}(x)$ is the sum of independent Bernoulli random variables with success probability at least $p^\star$. Thus, we have that $\mathbb{E}\left[\widetilde{N}(x)\right] \geq p^\star N(x)$ and multiplicative Chernoff's inequality gives

$$\mathbb{P}\left[\widetilde{N}(x) \leq \frac{1}{2} p^\star N(x)\right] \leq \exp\left(-\frac{1}{8} p^\star N(x)\right) \leq \exp\left(-\frac{1}{8} p^\star \tau_2\right).$$

Now, since $W$ is $(C, s)$-Zipfian, we have that $N(x) \leq \frac{CN}{r^s}$ for all $x \in \bigcup_i W_i$, so there can be at most $\left(\frac{CN}{\tau_2}\right)^{1/s}$ elements $x \in \bigcup_i W_i$ with $N(x) \geq \tau_2$. A union bound yields

$$\mathbb{P}\left[\exists x \in \bigcup_i W_i : N(x) \geq \tau_2, \widetilde{N}(x) < \frac{1}{2} p^\star N(x)\right] \leq \frac{\beta}{(CN)^{1/s}} \cdot \left(\frac{CN}{\tau_2}\right)^{1/s} \leq \beta,$$

which completes the proof. ∎

**Lemma C.4.** *For every dataset $W$, if the $\mathrm{WGM}$ is run with noise parameter $\sigma > 0$, threshold $T \geq 1$, and user contribution bound $\Delta_0 \geq 1$, then for every $\beta \in (0, 1)$, with probability at least $1 - \beta$ over $S \sim \mathrm{WGM}(W, \Delta_0)$, we have that $\widetilde{H}(x) \leq T_0, \forall x \in \widetilde{M}$, where $T_0 := T + \sigma \sqrt{2 \log\left(\frac{2N}{\beta}\right)}$ and $\widetilde{M} := \bigcup_i \widetilde{W}_i \setminus S$.*

*Proof.* By Line 4 in Algorithm 1, we have that for all $x \in \widetilde{M}$, its noisy weighted count is $\widetilde{H}'(x) < T$. Hence, by standard Gaussian concentration bounds (see Appendix A), with probability at least $1 - \beta$ over just the sampling of Gaussian noise in Line 3, we have $\widetilde{H}(x) \leq T_0$ for all $x \in \widetilde{M}$. ∎

We are now ready to prove Theorem 3.3.

*Proof.* (of Theorem 3.3) Let $\widetilde{W}$ be the random dataset obtained by sampling a set $\widetilde{W}_i$ of elements of size $\min\{\Delta_0, |W_i|\}$ without replacement from each $W_i$, and let $S$ be the overall output of the WGM. Let $\widetilde{N}(x) = \sum_{i=1}^n \mathbf{1}\{x \in \widetilde{W}_i\}$ be the frequency of item $x$ in the subsampled dataset and note that $\widetilde{N}(x) \leq \sqrt{q^\star} \widetilde{H}(x)$ for all $x \in \bigcup_i \widetilde{W}_i$, where $\widetilde{H}$ is the weighted histogram of item frequencies from $\widetilde{W}$. Let $\widetilde{M} = \bigcup_i \widetilde{W}_i \setminus S$ be the random variable denoting the set of items in $\bigcup_i \widetilde{W}_i$ but not in the algorithm's output $S$. Finally, define $\tau_1 := \frac{2\sqrt{q^\star} T_0}{p^\star}$ and $\tau_2 := \frac{8}{p^\star} \log\left(\frac{3(CN)^{1/s}}{\beta}\right)$, where

$$T_0 = T + \sigma\sqrt{2\log\left(\frac{2N}{\beta}\right)}.$$

Let $E_1$, $E_2$, and $E_3$ be the events of Lemma C.2, C.3, and C.4 respectively, setting the failure probability for each event to be $\frac{\beta}{3}$. Then, by the union bound, $E_1 \cap E_2 \cap E_3$ occurs with probability $1 - \beta$. It suffices to show that $E_1 \cap E_2 \cap E_3$ implies the stated upper bound on $\mathrm{MM}(W, S)$. We can decompose $\mathrm{MM}(W, S)$ into two parts, mass missed by subsampling and mass missed by noisy thresholding:

$$\mathrm{MM}(W, S) = \mathrm{MM}(W, \cup_i \widetilde{W}_i) + \frac{1}{N} \sum_{x \in \widetilde{M}} N(x). \tag{2}$$

Under $E_1$, we have that

$$\mathrm{MM}(W, \cup_i \widetilde{W}_i) \leq \frac{C^{1/s}}{s-1}\left(\frac{1}{p^\star N}\log\left(\frac{3(CN)^{1/s}}{\beta}\right)\right)^{\frac{s-1}{s}},$$

hence for the remainder of the proof, we will focus on bounding $\frac{1}{N}\sum_{x \in \widetilde{M}} N(x)$. First, we claim that under $E_2$ and $E_3$, we have that $N(x) \leq \max\{\tau_1, \tau_2\} =: \tau$ for all $x \in \widetilde{M}$. This is because, by event $E_3$, we have that for every $x \in \widetilde{M}$, $\widetilde{N}(x) \leq \sqrt{q^\star} T_0$. Thus, if there exists an $x \in \widetilde{M}$ such that $N(x) \geq \tau_2$, then by event $E_2$, it must be the case that $\frac{p^\star}{2} N(x) \leq \widetilde{N}(x) \leq \sqrt{q^\star} T_0$, which implies that $N(x) \leq \tau_1$.

Now, define $r_0 := \max\left\{\left(\frac{CN}{\tau}\right)^{1/s}, 1\right\}$. If $r \geq r_0$, then $\frac{CN}{r^s} \leq \tau$. Since $N(x) \leq \tau$ for every $x \in \widetilde{M}$, every such item has rank greater than $r_0$. Hence,

$$\frac{1}{N}\sum_{x \in \widetilde{M}} N(x) \leq \frac{1}{N}\sum_{r \geq r_0} N_{(r)} \leq \sum_{r \geq r_0} \frac{C}{r^s} \leq C\int_{r_0-1}^{\infty} t^{-s}dt \leq \frac{Cr_0^{1-s}}{s-1}.$$

Substituting in the definition of $r_0$ and continuing yields

$$\frac{1}{N}\sum_{x \in \widetilde{M}} N(x) \leq \frac{C^{1/s}}{s-1}\left(\frac{\tau}{N}\right)^{\frac{s-1}{s}} \leq \frac{C^{1/s}}{s-1}\left(\frac{\max\left\{\frac{2\sqrt{q^\star}T_0}{p^\star}, \frac{8}{p^\star}\log\left(\frac{3(CN)^{1/s}}{\beta}\right)\right\}}{N}\right)^{\frac{s-1}{s}}.$$

Now, we are ready to complete the proof. Using the decomposition of $\mathrm{MM}(W, S)$ in Equation 2 along with $E_1 \cap E_2 \cap E_3$ implies that

$$\mathrm{MM}(W, S) \leq \frac{C^{1/s}}{s-1}\left(\frac{1}{p^\star N}\log\left(\frac{3(CN)^{1/s}}{\beta}\right)\right)^{\frac{s-1}{s}} + \frac{C^{1/s}}{s-1}\left(\frac{\max\left\{2\sqrt{q^\star}T_0, 8\log\left(\frac{3(CN)^{1/s}}{\beta}\right)\right\}}{p^\star N}\right)^{\frac{s-1}{s}}$$

$$\leq \frac{C^{1/s}}{s-1}\left(\frac{1}{p^\star N}\right)^{\frac{s-1}{s}}\left(9\max\left\{\sqrt{q^\star}T_0, \log\left(\frac{3(CN)^{1/s}}{\beta}\right)\right\}\right)^{\frac{s-1}{s}}$$

$$= \frac{C^{1/s}}{s-1}\left(\frac{9}{p^\star N}\right)^{\frac{s-1}{s}}\max\left\{\sqrt{q^\star}\left(T + \sigma\sqrt{2\log\left(\frac{6N}{\beta}\right)}\right), \log\left(\frac{3(CN)^{1/s}}{\beta}\right)\right\}^{\frac{s-1}{s}}.$$

The proof is complete after noting that $\frac{\sqrt{q^\star}}{p^\star} = \frac{\max_i |W_i|}{\sqrt{q^\star}}$. $\blacksquare$

### C.2.3 PROOF OF THEOREM 3.6

As in the proof of Theorem 3.3, we start with the following lemma which bounds the maximum missing mass due to the subsampling step.

**Lemma C.5.** *Let $W$ be any dataset. Fix a user contribution bound $\Delta_0 \geq 1$. Let $\widetilde{W}$ be the random dataset such that $\widetilde{W}_i \subseteq W_i$ is a random sample without replacement of size $\min\{\Delta_0, W_i\}$. Then, for every $\beta \in (0, 1)$, with probability $1 - \beta$, we have that*

$$\mathrm{MM}_\infty(W, \cup_i \widetilde{W}_i) \leq \frac{\log\left(\frac{N}{\beta}\right)}{p^\star N}.$$

*where $p^\star := \min\left(1, \frac{\Delta_0}{\max_i |W_i|}\right)$.*

*Proof.* Let $W$ be any dataset, $\Delta_0 \geq 1$ and $\beta \in (0, 1)$. We will follow the same proof strategy as in the proof of Lemma C.2. Let $p_i := \min\left(1, \frac{\Delta_0}{|W_i|}\right)$ and $p^\star := \min_i p_i$. Fix an item $x \in \bigcup_i W_i$. Then,

$$\mathbb{P}\left[x \notin \bigcup_i \widetilde{W}_i\right] = \prod_{i:x \in W_i} (1 - p_i) \leq \exp\left(-\sum_{i:x \in W_i} p_i\right) \leq e^{-p^\star N(x)},$$

where $N(x) = \sum_i \mathbf{1}\{x \in W_i\}$. Fix $\beta \in (0, 1)$ and consider the threshold

$$Q := \frac{1}{p^\star} \log\left(\frac{N}{\beta}\right) \geq 1.$$

Since $\sum_{x \in \bigcup_i W_i} N(x) = N$, we have that that there are at most $\frac{N}{Q}$ items such that $N(x) > Q$. Hence, by the union bound, we get that

$$\mathbb{P}\left[\exists x \in \bigcup_i W_i \setminus \widetilde{W}_i \text{ and } N(x) > Q\right] \leq \frac{N}{Q} e^{-p^\star Q} \leq \beta.$$

Hence, with probability at least $1 - \beta$, we have that if $x \notin \bigcup_i \widetilde{W}_i$, then $N(x) \leq Q$, giving that

$$\mathrm{MM}_\infty(W, \cup_i \widetilde{W}_i) \leq \frac{\log\left(\frac{N}{\beta}\right)}{p^\star N},$$

which completes the proof. ∎

Now, we use Lemma C.5 to complete the proof of Theorem 3.6. Since the proof follows almost identically, we only provide a sketch here.

*Proof.* (sketch of Theorem 3.6) As in the proof of Theorem 3.3, define $q^\star = \min\{\max_i |W_i|, \Delta_0\}$ and $T_0 = T + \sigma\sqrt{2\log\left(\frac{6N}{\beta}\right)}$. Keep $\tau_1 := \frac{2\sqrt{q^\star T_0}}{p^\star}$ but take

$$\tau_2 := \frac{8}{p^\star} \log\left(\frac{3N}{\beta}\right).$$

Let $E_2$ be defined identically in terms of $T_0$. That is, $E_2$ is the event that $\widetilde{H}(x) \leq T_0$ for all $x \in \widetilde{M}$, where $T_0 = T + \sigma\sqrt{2\log\left(\frac{6N}{\beta}\right)}$ and $\widetilde{M} = \bigcup_i \widetilde{W}_i \setminus S$. Likewise, define $E_3$ in terms of $\tau_2$ analogous to that in the proof of Theorem 3.3. That is, $E_3$ is the event that for all $x \in \bigcup_i W_i$, either $N(x) < \tau_2$ or $\widetilde{N}(x) \geq \frac{p^\star}{2} \cdot N(x)$. The fact that $E_2$ occurs with probability at least $1 - \frac{\beta}{3}$ follows identitically from the proof of Theorem 3.3.. As for event $E_3$, note that we have

$$\mathbb{P}\left[\exists x \in \bigcup_i W_i : N(x) \geq \tau_2, \widetilde{N}(x) < \frac{1}{2}p^\star N(x)\right] \leq \frac{N}{\tau_2} \cdot e^{-\frac{p^\star \tau_2}{8}} \leq \frac{\beta}{3},$$

which follows similarly by using multiplicative Chernoff's, the union bound, and the fact that there can be at most $\frac{N}{\tau_2}$ items with frequency at least $\tau_2$. Hence, $E_3$ occurs with probability at least $1 - \frac{\beta}{3}$.

Then, by the union bound we have that with probability $1 - \frac{2\beta}{3}$, both $E_2$ and $E_3$ occur. When this happens, we have that $N(x) \leq \max\{\tau_1, \tau_2\}$ for all $x \in \widetilde{M}$ because either $N(x) \leq \tau_2$, or otherwise $\frac{1}{2}p^\star N(x) \leq \widetilde{N}(x) \leq \sqrt{q^\star}T_0$, implying that $N(x) \leq \tau_1$. Consequently, under $E_2$ and $E_3$, we have that

$$\max_{x \in \cup_i \widetilde{W}_i \setminus S} \frac{N(x)}{N} \leq \frac{\max\{\tau_1, \tau_2\}}{N}$$

By Lemma C.5, the event

$$\mathrm{MM}_\infty(W, \cup_i \widetilde{W}_i) \leq \frac{\log\left(\frac{3N}{\beta}\right)}{p^\star N}.$$

occurs with probability $1 - \frac{\beta}{3}$. Hence, under $E_1 \cap E_2 \cap E_3$, we have that

$$\mathrm{MM}_\infty(W, S) \leq \max\left\{\mathrm{MM}_\infty(W, \cup_i \widetilde{W}_i), \max_{x \in \widetilde{W} \setminus S} \frac{N(x)}{N}\right\}$$

$$\leq \frac{8}{p^\star N} \max\left\{\log\left(\frac{3N}{\beta}\right), \sqrt{q^\star}\left(T + \sigma\sqrt{2\log\left(\frac{6N}{\beta}\right)}\right)\right\},$$

which occurs with probability $1 - \beta$. Finally, plugging in $\sigma = \Theta\left(\frac{\sqrt{\ln(1/\delta)}}{\epsilon}\right)$ and $T = \tilde{\Theta}_{\Delta_0, \delta}(\sigma)$ completes the claim. ∎

### C.3 Proof of Theorem 4.3

Since the privacy guarantee follows by basic composition, we only focus on proving the utility guarantee in Theorem 4.3. First, we provide the proof of Lemma 4.2.

*Proof.* (of Lemma 4.2) Let $I = \mathcal{T}_k(W, D) \setminus S$ and $O = S \setminus \mathcal{T}_k(W, D)$. Then, $|I| = |O| \leq k$ and

$$\sum_{x \in \mathcal{T}_k(W, D)} N(x) - \sum_{x \in S} N(x) = \sum_{x \in I} N(x) - \sum_{x \in O} N(x).$$

Since $|I| = |O|$, there exists a one-to-one mapping $\pi : I \to O$ that pairs each item in $I$ with an item in $O$. Thus, we can write

$$\sum_{x \in I} N(x) - \sum_{x \in O} N(x) = \sum_{x \in I}(N(x) - N(\pi(x))).$$

By definition of $I$ and $O$, we have that for every $x \in I$ and $y \in O$, $\tilde{N}(y) \geq \tilde{N}(x)$. Hence, we have that $N(x) - N(y) \leq Z_y - Z_x$ and

$$\sum_{x \in I}(N(x) - N(\pi(x))) \leq \sum_{x \in I}(Z_{\pi(x)} - Z_x).$$

Define $R := \sum_{x \in I}(Z_{\pi(x)} - Z_x)$. Our goal is to get a high-probability upper bound on $R$ via concentration. By Gumbel concentration (Lemma A.2), with probability $1 - \beta$, we have that $\max_{x \in D} |Z_x| \leq \lambda \cdot \log(2|D|/\beta)$. Hence, under this event, we get that

$$R \leq 2k\lambda \cdot \log(2|D|/\beta).$$

Altogether, with probability $1 - \beta$, we have

$$\sum_{x \in \mathcal{T}_k(W, D)} N(x) - \sum_{x \in S} N(x) \leq R \leq 2k\lambda \cdot \log(2|D|/\beta).$$

Dividing by $N$ completes the proof. ∎

Combining Lemmas 4.2 and 4.1 then gives the following corollary.

**Corollary C.6.** *For every dataset $W$, domain $D$, $k \leq |D|$, $\epsilon > 0$, and $\delta \in (0,1)$, if Algorithm 3 is run with $\lambda = \tilde{\Theta}_\delta\left(\frac{\sqrt{k}}{\epsilon}\right)$ from Lemma 4.1, then Algorithm 3 is $(\epsilon, \delta)$-differentially private and with probability at least $1 - \beta$ over its output $S$, we have that*

$$\frac{1}{N}\left(\sum_{x \in \mathcal{T}_k(W,D)} N(x) - \sum_{x \in S} N(x)\right) \leq \tilde{O}_{\delta,\beta}\left(\frac{k^{3/2}\log|D|}{\epsilon N}\right).$$

With Corollary C.6 in hand, we are now ready to prove Theorem 4.3 after picking the same choice of $(\sigma, T)$ as in Theorem 3.2.

*Proof.* (of Theorem 4.3) Recall that by Theorem 3.6, if we set $\sigma = \Theta\left(\frac{\sqrt{\ln(1/\delta)}}{\epsilon}\right)$ and $T = \tilde{\Theta}_{\Delta_0,\delta/2}(\max\{\sigma, 1\})$, then the WGM is $(\epsilon/2, \delta/2)$-differentially private and with probability at least $1 - \beta/2$ over $D \sim \text{WGM}(W, \Delta_0)$, we have that

$$\text{MM}_\infty(W, D) \leq \tilde{O}_{\Delta_0,\delta/2,\beta/2}\left(\frac{\max_i |W_i|}{\epsilon N \sqrt{q^\star}}\right). \tag{3}$$

Let $\mathcal{T}_k(W)$ be the true set of top-$k$ elements and $\mathcal{T}_k(W, D)$ be the set of top-$k$ elements within the (random) domain $D$. Then, under this event, Equation 3 gives that

$$\frac{1}{N}\left(\sum_{x \in \mathcal{T}_k(W)} N(x) - \sum_{x \in \mathcal{T}_k(W,D)} N(x)\right) \leq k \cdot \text{MM}_\infty(W, D) \leq \tilde{O}_{\Delta_0,\delta/2,\beta/2}\left(\frac{k \cdot \max_i |W_i|}{\epsilon N \sqrt{q^\star}}\right). \tag{4}$$

By Corollary C.6, we know that running Algorithm 3 on input $W$, domain $D$ and $\lambda = \tilde{O}_{\delta/2}\left(\frac{\sqrt{k}}{\epsilon}\right)$ gives $(\epsilon/2, \delta/2)$-differentially privacy and that with probability at least $1 - \beta/2$, its output $S$ satisfies

$$\frac{1}{N}\left(\sum_{x \in \mathcal{T}_k(W,D)} N(x) - \sum_{x \in S} N(x)\right) \leq \tilde{O}_{\delta/2,\beta/2}\left(\frac{k^{3/2}\log|D|}{\epsilon N}\right). \tag{5}$$

Adding Inequalities 4 and 5 together and taking $|D| \leq |\bigcup_i W_i| =: M$ gives that with probability $1 - \beta$, the output $S$ of Algorithm 2 satisfies

$$\text{MM}^k(W, S) \leq \tilde{O}_{\beta,\delta,\Delta_0}\left(\frac{k}{N}\left(\frac{\max_i |W_i|}{\epsilon\sqrt{q^\star}} + \frac{\sqrt{k}\log(M)}{\epsilon}\right)\right),$$

which completes the proof. ∎

## C.4 PROOF OF THEOREM 4.5

Before we prove Theorem 4.5, we first present the pseudo-code (Algorithm 4) for the user peeling mechanism described in Section 4.2 along with its privacy and utility guarantees.

The following lemma gives the utility and privacy guarantee of Algorithm 4.

**Lemma C.7.** *For every dataset $W$, domain $D$, and $k \leq |D|$, if Algorithm 4 is run with noise parameter $\lambda > 0$, then with probability $1 - \beta$ over its output $S$, we have that*

$$\text{Hits}(W, S) \geq \left(1 - \frac{1}{e}\right)\text{Opt}(W, D, k) - 2k\lambda\log\left(\frac{2|D|k}{\beta}\right).$$

*where $\text{Opt}(W, D, k) := \arg\max_{S \subseteq D, |S| \leq k}\text{Hits}(W, S)$. If one picks $\lambda = \tilde{\Theta}_\delta\left(\frac{\sqrt{k}}{\epsilon}\right)$ from Lemma 4.1, then Algorithm 4 is $(\epsilon, \delta)$ differentially private and with probability $1 - \beta$ over its output $S$, we have*

$$\text{Hits}(W, S) \geq \left(1 - \frac{1}{e}\right)\text{Opt}(W, D, k) - \tilde{O}_{\delta,\beta}\left(\frac{k^{3/2}\log(|D|k)}{\epsilon}\right).$$

---

**Algorithm 4** User Peeling Mechanism

---

**Input:** Dataset $W$, domain $D$, number of elements $k$, noise-level $\lambda$

1  Initialize $W^1 \leftarrow W$, $D_1 \leftarrow D$, and output set $S_0 \leftarrow \emptyset$

2  **for** $j = 1, \dots, k$ **do**

3  $\quad$ Compute histogram $H^j(x) = \sum_i \mathbf{1}\{x \in W_i^j\}$ for all $x \in D_j$.

4  $\quad$ Compute noisy histogram $\tilde{H}^j(x) = H^j(x) + Z_x^j$ for all $x \in D_j$ where $Z_x^j \sim \mathrm{Gumbel}(\lambda)$.

5  $\quad$ Let $x_j \in \arg\max_{x \in D_j} \tilde{H}^j(x)$

6  $\quad$ Update $S_j \leftarrow S_{j-1} \cup \{x_j\}$, $D_{j+1} \leftarrow D_j \setminus \{x_j\}$, and $W^{j+1} \leftarrow \{W_i \in W^j : x_j \notin W_i\}$.

7  **end**

**Output:** $S_k$

---

Since Algorithm 4 also uses the peeling exponential mechanism, the privacy guarantee in Lemma C.7 follows exactly from Lemma 4.1, and so we omit the proof here. As for the utility guarantee, the proof is similar to the proof of Theorem 7 in Mitrovic et al. (2017). For the sake of completeness, we provide a self-contained analysis below.

*Proof.* Note that there exists at most $|D|k$ random variables $Z_x^j$. Let $E$ be the event that $|Z_x^j| \leq \alpha$ for all $x \in D$ and $j \in [k]$, where $\alpha = \lambda \ln\left(\frac{2|D|k}{\beta}\right)$. Then, by Gumbel concentration (Lemma A.2), we have that $\mathbb{P}(E) \geq 1 - \beta$. For the rest of this proof, we will operate under the assumption that event $E$ happens.

Define the function $f : 2^D \to \mathbb{N}$ as $f(S) := \sum_{i=1}^n \mathbf{1}\{W_i \cap S \neq \emptyset\} = \mathrm{Hits}(W, S)$. Then, $f$ is a monotonic, non-negative submodular function. For $x \in D$ and $S \subseteq D$ let $\Delta(x, S) := f(S \cup \{x\}) - f(S) \geq 0$. Let $S^\star \in \arg\max_{S \subseteq D, |S| \leq k} f(S)$ be the optimal subset of $D$ and denote $\mathrm{Opt} := f(S^\star)$. First, we claim that for every $S \subseteq D$,

$$\max_{x \in S^\star \setminus S} \Delta(x, S) \geq \frac{\mathrm{Opt} - f(S)}{k}. \tag{6}$$

This is because

$$\begin{aligned}
\mathrm{Opt} = f(S^\star) &\leq f(S^\star \cup S) \\
&\leq f(S) + \sum_{x \in S^\star \setminus S} \Delta(x, S) \\
&\leq f(S) + k \max_{x \in S^\star \setminus S} \Delta(x, S),
\end{aligned}$$

where the first two inequalities follows from monotonicity and submodularity respectively. Now, let $x_1, \dots, x_k$ be the (random) items selected by the algorithm, and $S_j = \{x_1, \dots, x_j\}$ with $S_0 = \emptyset$. At round $j \in [k]$, define the current domain $D_j := D \setminus S_{j-1}$. Let

$$x_j^\star \in \arg\max_{x \in S^\star \setminus S_{j-1}} \Delta(x, S_{j-1}).$$

Then, by Equation 6, we have $\Delta(x_j^\star, S_{j-1}) \geq \frac{\mathrm{Opt} - f(S_{j-1})}{k}$. This implies that

$$\begin{aligned}
\Delta(x_j, S_{j-1}) &\geq \frac{\mathrm{Opt} - f(S_{j-1})}{k} - (\Delta(x_j^\star, S_{j-1}) - \Delta(x_j, S_{j-1})) \\
&= \frac{\mathrm{Opt} - f(S_{j-1})}{k} - (H^j(x_j^\star) - H^j(x_j))
\end{aligned}$$

where the last equality stems from the fact that for every $x \in D_j$, $\Delta(x, S_{j-1}) = H^j(x)$.

Recall that $\tilde{H}^j(x) := H^j(x) + Z_x^j$ for all $x \in D$. We can upper bound $H^j(x_j^\star) - H^j(x_j)$ as

$$H^j(x_j^\star) - H^j(x_j) = (\tilde{H}^j(x_j^\star) - \tilde{H}^j(x_j)) + (Z_{x_j}^j - Z_{x_j^\star}^j) \leq Z_{x_j}^j - Z_{x_j^\star}^j,$$

where last inequality is because $\tilde{H}^j(x_j^\star) - \tilde{H}^j(x_j) \leq 0$ by the choice of $x_j \in \arg\max_{x \in D_j} \tilde{H}^j(x)$ and the fact that $x_j^\star \in S^\star \setminus S_{j-1} \subseteq D_j$. Therefore,

$$\Delta(x_j, S_{j-1}) \geq \frac{\mathrm{Opt} - f(S_{j-1})}{k} - \left(Z_{x_j}^j - Z_{x_j^\star}^j\right). \tag{7}$$

Let $G_j := \text{Opt} - f(S_j)$. Using $f(S_j) = f(S_{j-1}) + \Delta(x_j, S_{j-1})$, we rearrange Equation 7 to get

$$G_j \leq \left(1 - \tfrac{1}{k}\right) G_{j-1} + (Z_{x_j}^j - Z_{x_j^\star}^j).$$

On the event $E$, we have that $Z_{x_j}^j - Z_{x_j^\star}^j \leq |Z_{x_j}^j| + |Z_{x_j^\star}^j| \leq 2\alpha$, hence

$$G_j \; \leq \; \left(1 - \tfrac{1}{k}\right) G_{j-1} + 2\alpha.$$

Recursing for $j = 1, \ldots, k$ and using the fact that $G_0 = \text{Opt}$ and $(1 - \tfrac{1}{k})^k \leq e^{-1}$, gives

$$G_k \leq \frac{1}{e} \text{Opt} + 2\alpha k.$$

Substituting in the definition of $G_k$ and $\alpha = \lambda \ln\!\left(\tfrac{2|D|k}{\beta}\right)$ gives

$$f(S_k) \geq \left(1 - \tfrac{1}{e}\right) \text{Opt} - 2k\lambda \log\left(\tfrac{2|D|k}{\beta}\right).$$

■

The guarantee in Lemma C.7 is with respect to the optimal set of $k$ elements within the domain $D$. Since $D \subseteq \bigcup_i W_i$, we have that

$$\text{Opt}(W, D, k) \leq \text{Opt}(W, k).$$

The following simple lemma shows that when the domain $D$ contains high-frequency items from $W$, $\text{Opt}(W, D, k)$ is not too far away from $\text{Opt}(W, k)$.

**Lemma C.8.** *Fix a dataset $W$, $\tau \geq 0$, and let $D = \{x \in \bigcup_i W_i : N(x) \geq \tau\}$. Then,*

$$\text{Opt}(W, D, k) \geq \text{Opt}(W, k) - k\tau.$$

*Proof.* Recall that $f(S) := \text{Hits}(W, S)$ is a monotone, non-negative submodular function. Let $S_1 \subseteq \mathcal{X}$ be the subset of $\mathcal{X}$ that achieves $f(S_1) = \text{Opt}(W, k)$ and $S_2$ be the subset of $D$ that achieves $f(S_2) = \text{Opt}(W, D, k)$. By submodularity and the definition of $\mathcal{X} \setminus D$ we have that

$$f(S_1) \leq f(S_1 \cap D) + f(S_1 \cap \mathcal{X} \setminus D) \leq f(S_2) + k\tau,$$

which completes the proof. ■

Lemma C.8 allows us to use the $\text{MM}_\infty$ upper bound obtained by the WGM in Theorem 3.6 to upgrade the guarantee provided by Lemma C.7 to be in terms of $\text{Opt}(W, k)$ instead of $\text{Opt}(W, D, k)$. By using the same choice of $(\sigma, T)$ as in Theorem 3.2, we get the main result of this section. As before, since the privacy guarantee follows by basic composition, we only focus on proving the utility guarantee.

*Proof.* (of Theorem 4.5) Recall that by Theorem 3.6 that if we set $\sigma = \Theta\left(\frac{\sqrt{\ln(1/\delta)}}{\epsilon}\right)$ and $T = \tilde{\Theta}_{\Delta_0, \delta/2}(\max\{\sigma, 1\})$, then the WGM is $(\epsilon/2, \delta/2)$-differentially private and with probability at least $1 - \beta/2$ over $D \sim \text{WGM}(W, \Delta_0)$, we have that

$$\text{MM}_\infty(W, D) \leq \tilde{O}_{\Delta_0, \delta/2, \beta/2}\left(\frac{\max_i |W_i|}{\epsilon N \sqrt{q^\star}}\right).$$

By Lemma C.8, under this event we have that

$$\text{Opt}(W, D, k) \geq \text{Opt}(W, k) - \tilde{O}_{\Delta_0, \delta/2, \beta/2}\left(\frac{k \cdot \max_i |W_i|}{\epsilon \sqrt{q^\star}}\right).$$

Now, by Lemma C.7, we know that running Algorithm 4 on input $W$, domain $D$ and $\lambda = \tilde{O}_{\delta/2}\left(\frac{\sqrt{k}}{\epsilon}\right)$ gives $(\epsilon/2, \delta/2)$-differentially privacy and that with probability at least $1 - \beta/2$, its output $S$ satisfies

$$\text{Hits}(W, S) \geq \left(1 - \frac{1}{e}\right) \text{Opt}(W, D, k) - \tilde{O}_{\delta/2, \beta/2}\left(\frac{k^{3/2}}{\epsilon} \log\left(|D|k\right)\right)$$

Hence, by the union bound, with probability at least $1 - \beta$, both events occur and we have that

$$\text{Hits}(W, S) \geq \left(1 - \frac{1}{e}\right) \text{Opt}(W, D) - \tilde{O}_{\beta, \delta, \Delta_0}\left(\frac{k \cdot \max_i |W_i|}{\epsilon\sqrt{q^\star}} + \frac{k^{3/2}}{\epsilon}\log(Mk)\right),$$

where we use the fact that $|D| \leq M$. ∎

## D  LOWER BOUNDS

### D.1  LOWER BOUNDS FOR MISSING MASS

In this section, we prove Theorem 3.5. The following lemma will be useful.

**Lemma D.1.** *Let $\mathcal{A}$ be any $(\epsilon, \delta)$-differentially private algorithm satisfying Assumption 1. Then, for every dataset $W$ and item $x \in \bigcup_i W_i$, if $N(x) \leq \frac{1}{\epsilon}\ln\left(1 + \frac{e^\epsilon - 1}{2\delta}\right)$, then $\mathbb{P}_{S \sim \mathcal{A}(W)}[x \in S] \leq \frac{1}{2}$.*

*Proof.* Fix some dataset $W$. Let $\mathcal{A}$ be any randomized algorithm with privacy parameters $\epsilon \leq 1$ and $\delta \in (0, 1)$ satisfying Assumption 1. This means that for any item $x \in \bigcup_i W_i$ such that $N_W(x) = 1$, we have that

$$\mathbb{P}_{S \sim \mathcal{A}(W)}[x \in S] \leq \delta,$$

where for a dataset $W$, we define $N_W(x) := \sum_i \mathbf{1}\{x \in W_i\}$. Now, suppose $x \in \bigcup_i W_i$ is an item such that $N_W(x) = 2$. We can always construct a neighboring dataset $W'$ by removing a user such that $N_{W'}(x) = 1$. Thus, we have that

$$\mathbb{P}_{S \sim \mathcal{A}(W)}[x \in S] \leq e^\epsilon \mathbb{P}_{S \sim \mathcal{A}(W')}[x \in S] + \delta \leq \delta e^\epsilon + \delta = \delta(e^\epsilon + 1).$$

More generally, by unraveling the recurrence, we have that for any $x \in \bigcup_i W_i$ with $N_W(x) = b$,

$$\mathbb{P}_{S \sim \mathcal{A}(W)}[x \in S] \leq \delta \sum_{i=0}^{b-1} (e^\epsilon)^i = \delta \frac{e^{\epsilon b} - 1}{e^\epsilon - 1}.$$

Since $W$ was arbitrary, for any dataset $W$ and any $x \in \bigcup_i W_i$ with $N_W(x) = b$, we have that

$$\mathbb{P}_{S \sim \mathcal{A}(W)}[x \notin S] \geq 1 - \delta \frac{e^{\epsilon b} - 1}{e^\epsilon - 1}.$$

Now, consider the exclusion probability $p = 1/2$. Our goal is to compute the largest $b^\star$ such that for any dataset $W$, any $x \in \bigcup_i W_i$ with $N_W(x) \leq b^\star$ has $\mathbb{P}_{S \sim \mathcal{A}(W)}[x \notin S] \geq \frac{1}{2}$. It suffices to solve for $b$ in the inequality $1 - \delta\frac{e^{\epsilon b} - 1}{e^\epsilon - 1} \geq 1/2$, which yields $b \leq \frac{1}{\epsilon}\ln\left(1 + \frac{e^\epsilon - 1}{2\delta}\right) =: b^\star$. ∎

Lemma D.1 provides a uniform upper bound on the the probability that any private mechanism can output a low frequency item. We now use Lemma D.1 to complete the proof of Theorem 3.5.

*Proof.* (of Theorem 3.5) Let $b^\star = \frac{1}{\epsilon}\ln\left(1 + \frac{e^\epsilon - 1}{2\delta}\right)$ as in the proof of Lemma D.1. By Lemma D.1, for any dataset $W$ and item $x \in \bigcup_i W_i$ such that $N(x) \leq b^\star$, we have $\mathbb{P}_{S \sim \mathcal{A}(W)}[x \notin S] \geq \frac{1}{2}$. Consider a dataset $W^\star$ of size $n$, taking $n$ sufficiently large, where $\frac{N_{(r)}}{N} = \Theta\left(\frac{C}{r^s}\right)$ for all ranks $r \in [M^\star]$, where $M^\star = |\bigcup_i W_i^\star|$ (note that such a dataset is possible if, for example, one restricts each user to contribute exactly a single item). We can lower bound the missing mass of $\mathcal{A}$ on $W^\star$ as

$$\mathbb{E}_{S \sim \mathcal{A}(W^\star)}[\text{MM}(W^\star, S)] \geq \frac{1}{N} \sum_{x \in \bigcup_i W_i^\star, N(x) \leq b^\star} \mathbb{P}_{S \sim \mathcal{A}(W^\star)}[x \notin S] N(x)$$

$$\geq \frac{1}{2N} \sum_{x \in \bigcup_i W_i^\star, N(x) \leq b^\star} N(x).$$

Our next goal will be to find the smallest rank $r^\star$ such that $N_{(r^\star)} \leq b^\star$. It suffices to solve the inequality $\frac{CN}{r^s} \leq b^\star$ for $r$. Doing so gives that $r \geq \left\lceil \left(\frac{CN}{b^\star}\right)^{\frac{1}{s}} \right\rceil =: r^\star$. Hence, for this $W^\star$, we have that

$$
\begin{aligned}
\mathbb{E}_{S \sim \mathcal{A}(W^\star)}\left[\mathrm{MM}(W^\star, S)\right] &\geq \frac{1}{2N} \sum_{x \in \bigcup_i W_i^\star, N(x) \leq b^\star} N(x) \\
&= \frac{1}{2N} \sum_{r=r^\star}^{M^\star} N_{(r)} \\
&= \Omega\left(\frac{1}{N} \sum_{r=r^\star}^{M^\star} \frac{CN}{r^s}\right).
\end{aligned}
$$

Since $s > 1$, we can take $M^\star$ (and hence $n$) to be large enough so that

$$
\Omega\left(\frac{1}{N} \sum_{r=r^\star}^{M^\star} \frac{CN}{r^s}\right) = \Omega\left(\frac{1}{N} \int_{r^\star}^{\infty} \frac{CN}{r^s} dr\right).
$$

Thus,

$$
\begin{aligned}
\mathbb{E}_{S \sim \mathcal{A}(W^\star)}\left[\mathrm{MM}(W^\star, S)\right] &= \Omega\left(\frac{1}{N} \int_{r^\star}^{\infty} \frac{CN}{r^s} dr\right) \\
&= \Omega\left(\frac{C}{s-1}(r^\star)^{1-s}\right) \\
&= \Omega\left(\frac{C^{1/s} N^{(1-s)/s}}{s-1}(b^\star)^{(s-1)/s}\right) \\
&= \Omega\left(\frac{C^{1/s} N^{-(s-1)/s}}{s-1}\left(\frac{1}{\epsilon}\ln\left(1 + \frac{e^\epsilon - 1}{2\delta}\right)\right)^{(s-1)/s}\right) \\
&= \Omega\left(\frac{C^{1/s}}{s-1} \cdot \left(\frac{1}{\epsilon N}\right)^{(s-1)/s} \cdot \ln\left(1 + \frac{e^\epsilon - 1}{2\delta}\right)^{(s-1)/s}\right),
\end{aligned}
$$

which completes the proof. ∎

## D.2 LOWER BOUNDS FOR TOP-$k$ SELECTION

*Proof.* (of Corollary 4.4) Define $b^\star = \frac{1}{\epsilon}\ln\left(1 + \frac{e^\epsilon - 1}{2\delta}\right)$ like in Lemma D.1. Recall from Lemma D.1, that for any dataset $W$ and any item $x \in \bigcup_i W_i$ such that $N(x) \leq b^\star$, we have that

$$
\mathbb{P}_{S \sim \mathcal{A}(W,k)}\left[x \notin S\right] \geq \frac{1}{2}.
$$

Consider any dataset $W^\star$ such that $|\bigcup_i W_i^\star| = k$ and $N_{(i)} = b^\star$ for all $i \in [k]$. Then $\sum_{i=1}^k N_{(i)} = kb^\star$. But, we also have that $\mathbb{E}_{S \sim \mathcal{A}(W,k)}\left[\sum_{x \in S} N(x)\right] \leq \frac{kb^\star}{2}$. Hence,

$$
\mathbb{E}_{S \sim \mathcal{A}(W^\star, k)}\left[\mathrm{MM}^k(W^\star, S)\right] \geq \frac{kb^\star}{2N} = \tilde{\Omega}_\delta\left(\frac{k}{\epsilon N}\right),
$$

completing the proof. ∎

## D.3 LOWER BOUNDS FOR $k$-HITTING SET.

*Proof.* (of Corollary 4.6) Define $b^\star = \frac{1}{\epsilon}\ln\left(1 + \frac{e^\epsilon - 1}{2\delta}\right)$ as in Lemma D.1. Recall from Lemma D.1, that for any dataset $W$ and any item $x \in \bigcup_i W_i$ such that $N(x) \leq b^\star$, we have that

$$
\mathbb{P}_{S \sim \mathcal{A}(W,k)}\left[x \notin S\right] \geq \frac{1}{2}.
$$

This is due to our restriction that $\mathcal{A}(W, k) \subseteq \bigcup_i W_i$. Consider any dataset $W^\star$ consisting of $k$ unique items and $n = kb^\star$ users such that each item hits a disjoint set of $b^\star$ users. Since the frequency of each of the $k$ items is at most $b^\star$, $\mathcal{A}$ can output each item with probability at most $1/2$. Hence, in expectation, $\mathcal{A}$ outputs at most $k/2$ distinct items, hitting at most $\frac{kb^\star}{2}$ users, while the optimal set of items includes all $k$ items and hits all users. ∎

## E    EXPERIMENT DETAILS

In this section, we provide details about the datasets used in Section 5. Table 1 provides statistics for the 6 datasets we consider. Reddit (Gopi et al., 2020) is a text-data dataset of posts from r/askreddit. For this dataset, each user corresponds to a set of documents, and following prior methodology, we take a user's item set to be the set of tokens used across all documents. Movie Reviews (Harper & Konstan, 2015) is a dataset containing movie reviews from the MovieLens website. Here, we group movie reviews by user-id, and take a user's itemset to be the set of movies they reviewed. Amazon Games, Pantry, and Magazine (Ni et al., 2019) are review datasets for video games, prime pantry, and magazine subscriptions respectively. Like for Movie Reviews, we group rows by user-id and take a user's item set to the set of items they reviewed. Finally, Steam Games (Steam, 2025) is a dataset of 200k user interactions (purchases/play) on the Steam PC Gaming Hub. As before, we group the rows by user-id and take a user's itemset to the be the set of games they purchased/played.

| Dataset | No. Users | No. Items | No. Entries |
|---|---|---|---|
| Reddit | 245103 | 631855 | 18272211 |
| Movie Reviews | 162541 | 59047 | 25000095 |
| Amazon Games | 1540618 | 71982 | 2489395 |
| Steam Games | 12393 | 5155 | 128804 |
| Amazon Magazine | 72098 | 2428 | 88318 |
| Amazon Pantry | 247659 | 10814 | 447399 |

Table 1: Number of users, items, and entries (user-item pairs) for each dataset

### E.1    RANK VS. FREQUENCY PLOTS

In order to get meaningful upper bounds on MM, Theorem 3.3 requires that $W$ be $(C, s)$-Zipfian for $s > 1$. Figures 4 and 5 present log-log plots of frequency vs. rank for the large and small datasets respectively. In all cases, we observe that the real-world datasets we consider are $(C, s)$-Zipfian for $s > 1$ and sufficiently large $C$. Note that our definition of a Zipfian dataset only requires the frequency vs. rank plot to be *upper bounded* by a decaying polynomial.

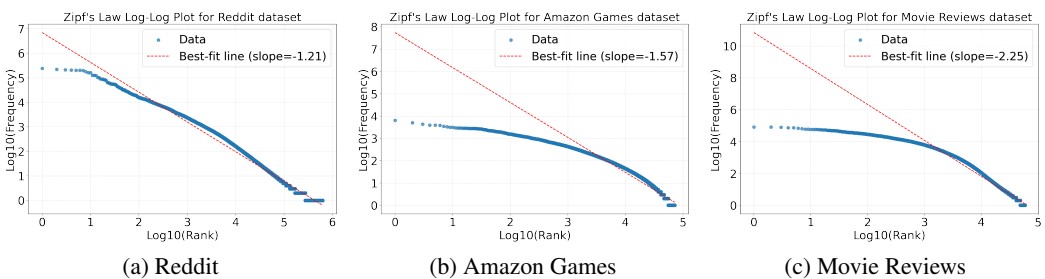

(a) Reddit          (b) Amazon Games          (c) Movie Reviews

Figure 4: Log-log plot of frequency vs. rank for large datasets

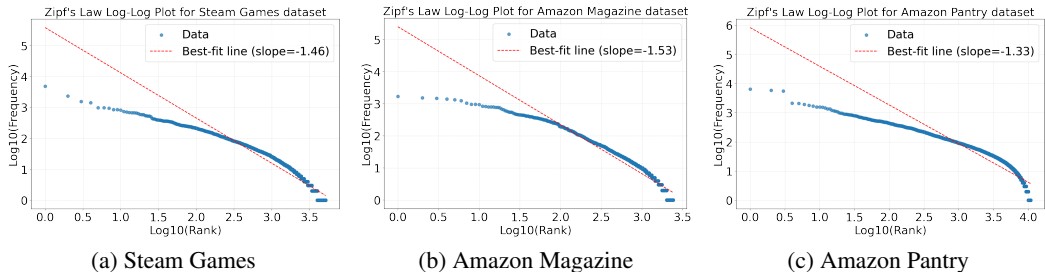

(a) Steam Games        (b) Amazon Magazine        (c) Amazon Pantry

Figure 5: Log-log plot of frequency vs. rank for small datasets

## E.2 USER ITEM SET SIZE DISTRIBUTIONS

Figures 6 and 7 plot the Empirical CDF (ECDF) of user item set sizes for the large and small datasets respectively.

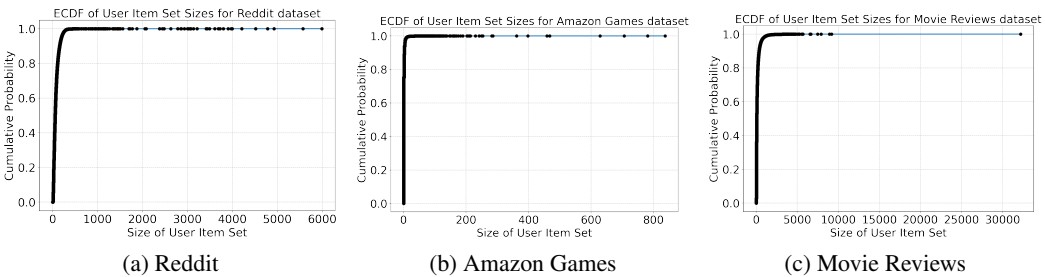

(a) Reddit        (b) Amazon Games        (c) Movie Reviews

Figure 6: ECDFs of user item set sizes for large datasets.

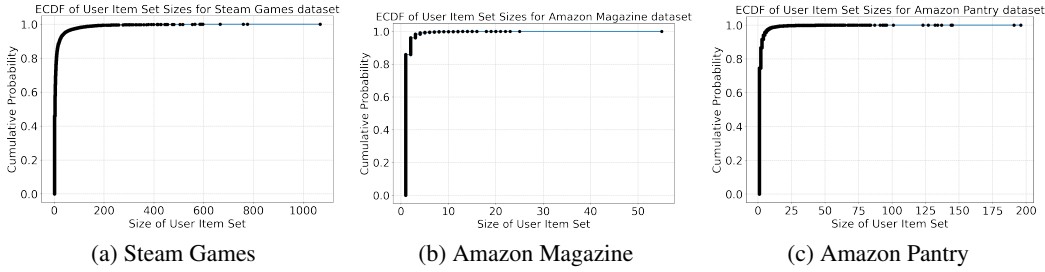

(a) Steam Games        (b) Amazon Magazine        (c) Amazon Pantry

Figure 7: ECDFs of user item set sizes for small datasets.

## F ADDITIONAL EXPERIMENTAL RESULTS

### F.1 PRIVATE DOMAIN DISCOVERY

#### F.1.1 RESULTS FOR SMALL DATASETS

Figure 8 plots the MM as a function of $\Delta_0$ for the small datasets. Again, we find that the WGM achieves comparable performance to the policy mechanism while being significantly more computationally efficient.

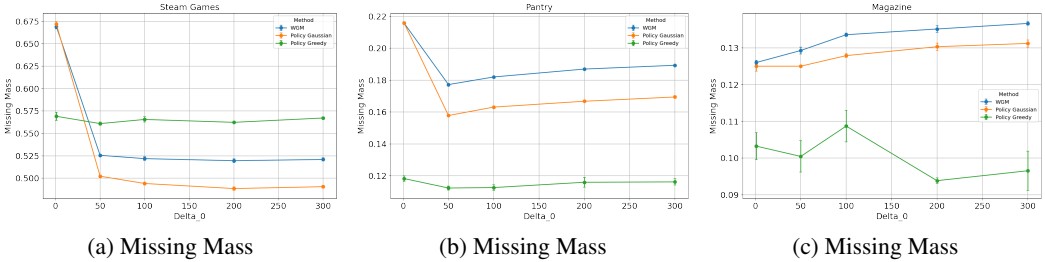

(a) Missing Mass        (b) Missing Mass        (c) Missing Mass

Figure 8: MM as a function of $\Delta_0 \in \{1, 50, 100, 150, 200, 300\}$ for the small datasets.

### F.1.2 RESULTS FOR $\epsilon = 0.10$

Figures 9 and 10 plot the MM for the large and small datasets respectively when $\epsilon = 0.1$ and $\delta = 10^{-5}$.

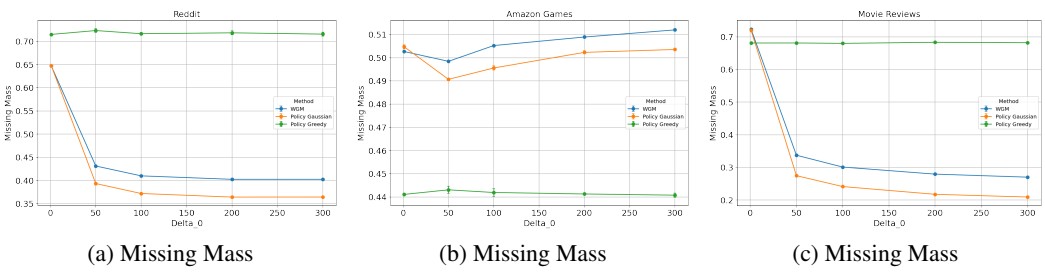

(a) Missing Mass        (b) Missing Mass        (c) Missing Mass

Figure 9: MM as a function of $\Delta_0 \in \{1, 50, 100, 150, 200, 300\}$ for large datasets when $\epsilon = 0.1$ and $\delta = 10^{-5}$.

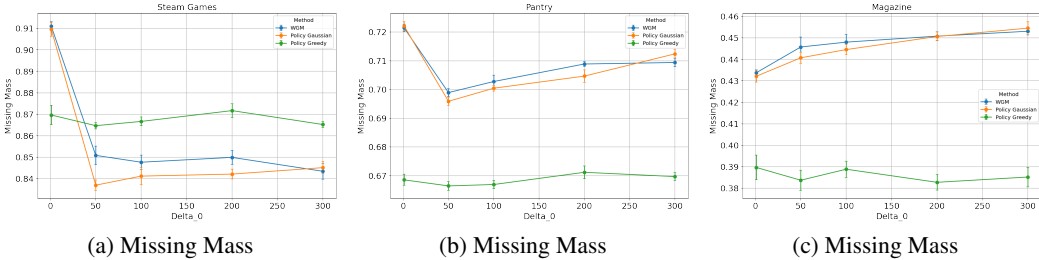

(a) Missing Mass        (b) Missing Mass        (c) Missing Mass

Figure 10: MM as a function of $\Delta_0 \in \{1, 50, 100, 150, 200, 300\}$ for small datasets when $\epsilon = 0.1$ and $\delta = 10^{-5}$.

### F.2 TOP-$k$ SELECTION

The top-$k$ $\ell_1$ loss is defined as

$$\ell_1^k(W, S) = \sum_{i=1}^{\min\{|S|,k\}} |N_{(i)} - N(S_i)| + \sum_{i=\min\{|S|,k\}}^{k} N_{(i)},$$

where $S$ is any *ordered* sequence of items. Unlike the top-$k$ MM, the top-$k$ $\ell_1$ loss cares about the order of items output, and hence is a more stringent measure of utility.

Figure 11 plots the top-$k$ $\ell_1$ MM for $\epsilon = 1.0$, $\delta = 10^{-5}$, and $\Delta_0 = 100$. Similar to Figure 2, we observe that our method (purple) achieves significantly less Top-$k$ $\ell_1$ loss compared to all baselines.

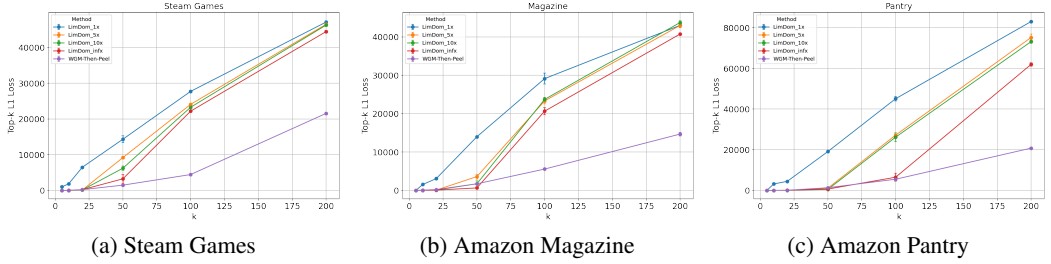

(a) Steam Games    (b) Amazon Magazine    (c) Amazon Pantry

Figure 11: Top-$k$ $\ell_1$ loss vs. $k$ with $\epsilon = 1.0$, $\delta = 10^{-5}$, and $\Delta_0 = 100$.

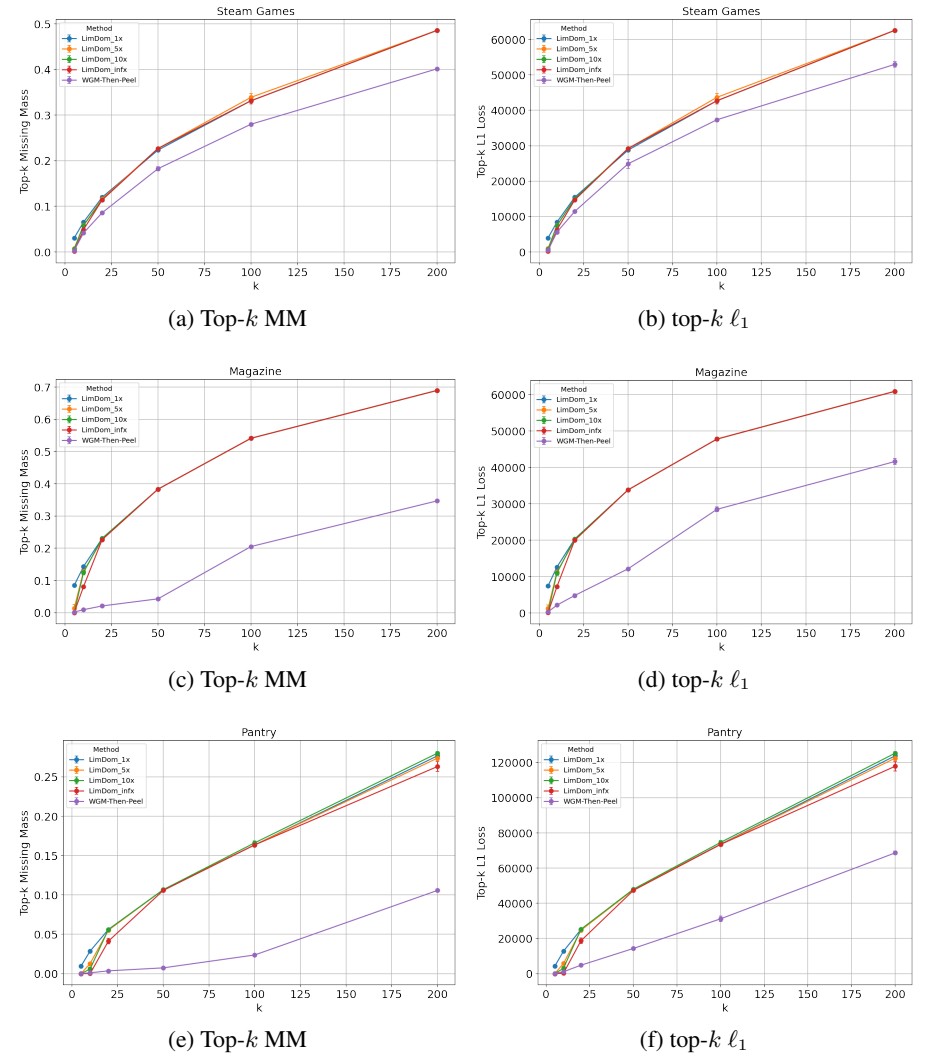

(a) Top-$k$ MM

(b) top-$k$ $\ell_1$

(c) Top-$k$ MM

(d) top-$k$ $\ell_1$

(e) Top-$k$ MM

(f) top-$k$ $\ell_1$

Figure 12: Top-$k$ MM and Top-$k$ $\ell_1$ vs. $k$ for $k \in \{5, 10, 20, 50, 100, 200\}$, with $\epsilon = 0.1$, $\delta = 10^{-5}$ and $\Delta_0 = 100$.

Figure 12 plots the top-$k$ MM and top-$k$ $\ell_1$ loss for the small datasets when $\epsilon = 0.1$ and $\delta = 10^{-5}$. Like the case when $\epsilon = 1.0$, our method (purple) continues to outperform the baselines across all $k$ values.

## F.3 $k$-HITTING SET

Figure 13 plots the Number of missed users for the small datasets when $\epsilon = 0.1$, $\delta = 10^{-5}$, and $\Delta_0 = 100$. We observe that our method (blue) performs comparably and sometimes outperforms the case where the domain $\bigcup_i W_i$ is public.

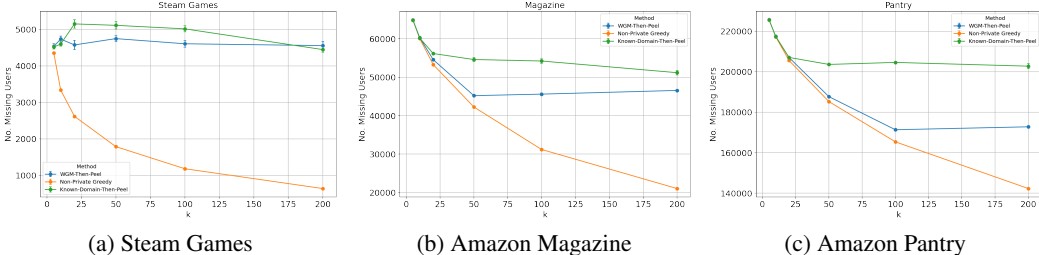

| (a) Steam Games | (b) Amazon Magazine | (c) Amazon Pantry |

Figure 13: Number of missed users vs. $k$ for $k \in \{5, 10, 20, 50, 100, 200\}$ with $\epsilon = 0.1, \delta = 10^{-5}$, and $\Delta_0 = 100$.

