# OpenReview forum: "Differentially Private Domain Discovery"
_ICLR.cc/2026/Conference — ICLR 2026 Oral_

### Official Review · Reviewer_Ed3k · 2025-10-31

**Soundness:** 3
**Presentation:** 3
**Contribution:** 2
**Rating:** 6
**Confidence:** 3

**Summary:**

This paper studies differentially private domain discovery, where each user holds a subset of items from an unknown shared domain, and the goal is to output an informative subset while preserving privacy. The authors revisit the Weighted Gaussian Mechanism (WGM) and show that, when utility is measured in terms of missing mass (the fraction of unrecovered probability mass rather than set cardinality), WGM achieves strong and in some cases near-optimal guarantees.

Key contributions include:

1. Reformulating the DP set union problem in terms of mass instead of cardinality, and proving that WGM achieves near-optimal ℓ1 missing mass on Zipfian data, and a distribution-free  ℓ∞ missing mass bound.
2. Extending the analysis to unknown-domain variants of top-k and k-hitting set, showing new utility guarantees by using WGM as a preprocessing step.
3. Conducting experiments on six real-world datasets, demonstrating that WGM-based methods outperform existing baselines in accuracy and scalability.

**Strengths:**

1.  **Originality**: Reformulating domain discovery in terms of missing mass provides a novel and tractable utility metric.
2.  **Quality**: The analysis is rigorous, with clear theorems and well-motivated assumptions. The near-optimal ℓ1 and distribution-free ℓ∞ guarantees add strong theoretical credibility.
3.  **Clarity**: Well-written, clear notation, and logically structured.
4.  **Significance**: The results contribute a unified framework for private domain discovery, potentially influencing future research in private data analytics and unknown-domain learning.

**Weaknesses:**

1.  The Zipfian assumption may restrict generality; robustness under other distributions is not fully discussed.
2.  Section 6 outlines directions but lacks concrete proposals (e.g., how adaptive subsampling might integrate into WGM).

**Questions:**

1.  How does WGM perform on non-Zipfian or adversarial data?
2.  How does the mechanism perform under correlated user data?
3.  Does adding Gaussian noise per element remain efficient when domain size is very large?

---

> ### Author Response · Authors · 2025-11-19
>
> Thanks for the review! We've attempted to address the most salient comments below. Please let us know if further questions arise.
>
> > The Zipfian assumption may restrict generality; robustness under other distributions is not fully discussed.
>
> As noted in Lines 132-141, without placing any distributional assumptions, it is not possible to derive meaningful bounds on the MM. That said, we agree that there may be other distributional assumptions that could be of broader interest. We leave this for future work.
>
> > How does WGM perform on non-Zipfian or adversarial data?
>
> As noted in Lines 132-141 one cannot do better than $1- \delta$ for an adversarial dataset (i.e. the dataset where all items appear exactly once). That said, our experiments evaluate and show the robustness of the WGM on six real-world datasets.
>
> > How does the mechanism perform under correlated user data?
>
> Our theoretical guarantees make no assumptions about user correlations and only rely on the item frequency histogram exhibiting Zipf's law. So, user data can be correlated as long as Zipf's law is obeyed. Note that in practice, one does not need the dataset to be Zipfian in order to run the WGM.
>
> > Does adding Gaussian noise per element remain efficient when domain size is very large?
>
> Yes. First, WGM only requires computing a noisy count for each element actually present in the underlying dataset. Second, Gaussian noise is easy to parallelize, as the noise added to each count is iid.

---

### Official Review · Reviewer_sdpD · 2025-10-31

**Soundness:** 3
**Presentation:** 3
**Contribution:** 2
**Rating:** 6
**Confidence:** 4

**Summary:**

The paper studies the problem of discovering items from an unknown domain in a *differentially private* manner, under a user-level privacy guarantee.

The main technical tool is the Weighted Gaussian Mechanism (WGM) (Gopi et al., 2020), which operates as follows:
1. It builds a histogram over items held by all clients, normalizing each user’s contribution so that its $\ell_2$-norm is bounded by $1$.
2. It adds Gaussian noise to the non-zero entries of the histogram.
3. It applies a thresholding step to remove low-frequency items.

The paper analyzes the performance of WGM under two error metrics—the $\ell_1$ and $\ell_\infty$ norms of the *missing mass*—where the analysis for the $\ell_1$ case assumes a Zipfian distribution over the domain.

Finally, the authors extend their approach to the top-$k$ discovery and $k$-hitting set problems, and provide empirical evaluations demonstrating the effectiveness of their method.

**Strengths:**

The problem studied and the proposed error metrics are interesting.

It is also good to see a systematic and thorough investigation of this topic, which helps clarify the behavior of differentially private item discovery mechanisms under various conditions.

**Weaknesses:**

1. Some key related works are missing and should be discussed.

2. The motivation and interpretation of the experimental design are less clear:
   * It is unclear whether the datasets used in the experiments are justified to follow the Zipfian distribution assumed in the analysis—particularly for parameters $s > 1$ (and not even $s = 1$).

   * It would be helpful to clarify whether the key theoretical results, such as **Theorem 3.3** and **Corollary 3.4**, are empirically validated in the experiments.

**Questions:**

1. It would be helpful to clarify whether the error metrics defined in **Definition 2.2** are newly proposed in this paper or have appeared in prior work.
   If they are not novel, please include appropriate citations to previous literature.

2. The asymptotic upper and lower bounds for differentially private top-$k$ selection are already well studied. For instance:
   * Bafna, Mitali, and Jonathan Ullman. *“The Price of Selection in Differential Privacy.”* In *Conference on Learning Theory*, pp. 151–168. PMLR, 2017.
   * Steinke, Thomas, and Jonathan Ullman. *“Tight Lower Bounds for Differentially Private Selection.”* In *FOCS 2017*, pp. 552–563. IEEE, 2017.

   It is known that the cumulative gap of $\tilde{O}(k^{3/2})$ between the true and selected top-$k$ scores is tight.
   Therefore, it may not be necessary to
   * re-prove **Lemma 4.2**, or
   * include **Corollary 4.4**, whose lower bound is not tight.

3. The citation for **Lemma 4.1** appears to be inaccurate.
   The equivalence between adding Gumbel noise and iteratively applying the exponential mechanism (without replacement) was shown as early as Durfee & Rogers, NeurIPS 2019.
   Moreover, their updated arXiv version (see footnote on p.11 of [arXiv:1905.4273](https://arxiv.org/pdf/1905.4273)) acknowledges that this technique was proposed even earlier.

4. In **Theorem 3.2**, it would be clearer to express $T$ in $\Theta$-notation rather than $\tilde{\Theta}$-notation, since
   (1) $\sigma$ is already expressed in $\Theta$-notation, and
   (2) it would make the asymptotic behavior of $T$ easier to interpret.

5. The text in **Figures 1–3** is too small to read when printed. Please consider increasing the font size for readability.

---

> ### Author Response · Authors · 2025-11-19
>
> Thanks for the review! We've attempted to address the most salient comments below. Please let us know if further questions arise.
>
> > It is unclear whether the datasets used in the experiments are justified to follow the Zipfian distribution assumed in the analysis—particularly for parameters  $s > 1$ (and not even $s=1$).
>
> The updated submission draft includes log-log plots of frequency vs. rank for the different datasets in Appendix E.1. These demonstrate that, although they were not chosen for this property, these datasets are indeed $(C, s)$-Zipfian for $s > 1$. We also note that the Zipfian assumption is required only for our utility analysis, not the privacy analysis.
>
> > It would be helpful to clarify whether the key theoretical results, such as Theorem 3.3 and Corollary 3.4, are empirically validated in the experiments.
>
> Theorem 3.3 shows that the missing mass decays proportionally to $\max_i |W_i|/N$. In Figure 8 (of the updated manuscript), we find that the missing mass on the Steam Games dataset is significantly higher than that for Amazon Magazine and Pantry. After computing $\max_i |W_i|/N$ for each of these three datasets, we find that $\max_i |W_i|/N$ for the Steam Games dataset is an order of magnitude larger than that for Amazon Magazine and Pantry. In addition, Theorems 4.3 and 4.5 indicate that top-$k$ MM and the number of missed users should increase with $k$. This is corroborated empirically by Figures 2 and 3.
>
> > It would be helpful to clarify whether the error metrics defined in Definition 2.2 are newly proposed in this paper or have appeared in prior work. If they are not novel, please include appropriate citations to previous literature.
>
> Missing mass is a classical idea from statistics, but to the best of our knowledge, its specific application to private set union is new to our work.

---

> > ### Comment · Reviewer_sdpD · 2025-11-19
> >
> > Regarding Questions 2 and 3: What would be the future plans for Lemma 4.2 and Corollary 4.4? Also, how about the citation for Lemma 4.1?

---

> > > ### Author Response · Authors · 2025-11-20
> > >
> > > Regarding the citation for Lemma 4.1, we will make sure to update it to the one you suggested in the final version. As for the plans for Lemma 4.2 and Corollary 4.4, we agree that the upper bound in Lemma 4.2 is essentially known as it is just a straightforward application of Gumbel concentration inequalities. We will make sure to highlight that and include citations, but for completeness sake, we think it is useful to still include its proof in the Appendix. As for Corollary 4.4,  we found that the lower bounds in the papers you linked are slightly different than the one we have. Namely, in these papers, one proves a lower bound for algorithms with an accuracy guarantee for a distribution over datasets. In our work, we prove lower bounds for algorithms with the constraint that it is only allowed to output items they have seen in their input. As such, we are able to give a single dataset, where all algorithms with this property fail. We feel that due to this subtle difference and for the sake of completeness, it is best to keep Corollary 4.4 as is. That said, we will make sure to highlight and cite these existing lower bounds for top-k selection

---

### Official Review · Reviewer_FKQB · 2025-11-01

**Soundness:** 3
**Presentation:** 3
**Contribution:** 3
**Rating:** 6
**Confidence:** 4

**Summary:**

This paper studies several questions about user-level differentially private (DP) algorithms on an unknown domain. The dataset consists of users, each with a subset $W_i$ of items from an unknown (or enormous) universe $\mathcal{X}$ and the goal is to output a set $S$ with large overlap with the union $\bigcup_i W_i$ while maintaining DP with respect to adding/removing any given subset $W_i$. For instance, in the set union problem, the goal is to output $S$ which is a subset of the union with as large cardinality as possible. Unfortunately, for this problem, there are no known absolute utility bounds: prior works can compare algorithms but have no guarantees on the size of the output. This paper investigates a relaxed objective which is missing mass. Consider the empirical distribution over the universe given by the input dataset (the multiset union of all $W_i$) and call this vector $f$. The missing mass is defined as the $\ell_1$ norm of $ f_{\mathcal{X} \setminus S}$, the fraction of elements not contained in $S$. The authors note that $\ell_0$ norm is the normal private set union objective and the $\ell_p$ missing mass for $p > 1$ may also be of interest.

The first main result is about set union with $\ell_1$ missing mass. The authors show that the standard weighted Gaussian mechanism (WGM) from prior work achieves a missing mass guarantee if the item frequencies $f$ follow a power law (Zipfian distribution). With a Zipfian exponent $s$, a simplified version of the bound is $\tilde{O}\left(\left(\frac{\max_i |W_i|}{\epsilon N \sqrt{\Delta_0}}\right)^{(s-1)/s}\right)$ where $\Delta_0$ is the standard "contribution cap": in a preprocessing step, each $W_i$ is subsampled to keep only $\Delta_0$ elements.
Furthermore, the authors show that the $\left(\frac{1}{\epsilon N}\right)^{(s-1)/s}$ dependence is necessary for any private algorithm.
The authors show a simpler result on the $\ell_\infty$ missing mass without any Zipfian assumptions. This amounts to showing that there is a frequency threshold, above which, any item will be returned by WGM.

The authors show how to use the $\ell_\infty$ analysis to get utility bounds for top-$k$ selection and $k$-hitting set on unknown (very large) domains by first applying set union and then running a known domain algorithm. They use the $\ell_\infty$ missing mass bound from set union to get missing mass type utility bounds for top-$k$ and approximation algorithm bounds for $k$-hitting set.

**Strengths:**

- Achieving absolute utility bounds for private set union has been a challenge in the existing literature. By relaxing the objective to missing mass as opposed to just the cardinality of $S$ as well as introducing an assumption on the frequency distribution, the authors obtain absolute bounds on the utility of the standard WGM algorithm.

- The upper and lower bounds on Zipfian data show that different algorithmic ideas cannot significantly improve upon the basic WGM algorithm in this setting (at least in terms of missing mass).

- The authors show how these bounds can be translated to problems where set union can be used as a subroutine for the top-$k$ and $k$-hitting set problems.

- One interesting facet of the theoretical analysis is that the preprocessing step, where sets $W_i$ are subsampled to have a maximum cardinality cap, is an important factor in the bounds. In prior studies of private set union, the details of step are not thoroughly investigated.

**Weaknesses:**

An asymptotic bound on the missing mass of a mechanism in a specific setting would be more useful if it had implications in practice either by predicting practical performance in parameter settings or by allowing different algorithms to be compared theoretically.

- The experimental results could be significantly more in depth: this would highlight several aspects of the theoretical work. Some concrete suggestions:
  - It would be instructive to compare the theoretical error bound against empirical performance on synthetic Zipfian data.
  - It would be useful to plot the frequency distribution on the real datasets to see if they follow a power law.
  - Reporting the size distribution of $|W_i|$ would be interesting to compare the setting of $\Delta_0$ to those statistics. The theoretical result suggests that $\Delta_0$ should be set to $\max |W_i|$ though increasing $\Delta_0$ worsens performance for the Amazon games dataset and the small datasets. Is this because $\Delta_0$ has exceeded the size of any $W_i$ or for another reason?
- The statement that the gap between WGM and the policy baselines is smaller in the missing mass case compared to the $\ell_0$ cardinality case is misleading. While the gap is 5 percentage points (MM is measured as a percentage), WGM performs up to 50% worse than the baselines in some instances when measured as a ratio. This is the apples-to-apples comparison to make when commenting that "for cardinality... sequential methods often output $\approx 2X$ more items [than WGM]."
- Comparison to more baseline algorithms would also be interesting. In the theoretical results, it is very reasonable to only analyze WGM as it is a well-studied algorithm and other candidate algorithms may be very difficult to analyze. On the other hand, if an important empirical takeaway is that "WGM obtains MM within 5% of that of the policy mechanisms, in spite of their significantly more intensive computation," it would be useful to compare to other methods of Swanberg et al. and Chen et al. which offer better cardinality results while still using a similar amount of computation to WGM.
- Missing mass analysis of any other algorithm, for example the weighted Laplace mechanism to start, would be interesting.

**Questions:**

- Consider the statement "Note that in Theorem 3.3 the missing mass decays as the total number of items N grows. Moreover, as C decreases or s increases, the upper bound on missing mass decreases." Should the second sentence be conditional on $N$ being sufficiently large compared to $T$ and $\sigma$ (which should be the case for reasonable privacy parameter settings).

- An interesting direction would be to show that WGM is close to optimal on Zipfian data for all $\ell_p$ missing mass for $p > 1$. Is there any chance of showing something like this? Similarly, plotting missing mass as $p$ increases would be interesting, especially if the plots were significantly different for different algorithms.

- Are both changes of (1) considering missing mass and (2) Zipfian data necessary for giving an absolute utility guarantee? Is there a specific technical challenge to analyzing the cardinality of the set output by WGM on Zipfian data?

---

> ### Author Response · Authors · 2025-11-19
>
> Thanks for the review! We've attempted to address the most salient comments below. Please let us know if further questions arise.
>
> > It would be instructive to compare the theoretical error bound against empirical performance on synthetic Zipfian data.
>
> We suggest that adding new experiments on new datasets may be too significant a change to make during the response period. However, we are happy to add material to better contextualize the existing experiments (see responses immediately below).
>
> > It would be useful to plot the frequency distribution on the real datasets to see if they follow a power law.
>
> We have uploaded a new version of the manuscript with log-log plots of frequency vs. rank in Appendix E.1. From these, we observe that the datasets that we use for our MM experiments are indeed $(C, s)$-Zipifian for $s > 1$.
>
> > Reporting the size distribution of $|W_i|$ would be interesting to compare the setting of $\Delta_0$ to those statistics. The theoretical result suggests that $\Delta_0$ should be set to $\max |W_i|$ though increasing $\Delta_0$ worsens performance for the Amazon games dataset and the small datasets. Is this because $\Delta_0$ has exceeded the size of any $W_i$ or for another reason?
>
> We have uploaded a new version of the manuscript with histograms of user item set sizes for each dataset in Appendix E.2.  As suggested, for the Amazon Games dataset, we observe that the vast majority of users have a user item set size smaller than 50. Our theoretical guarantees indicate that in this regime, MM should degrade as $\Delta_0$ increases, and this is what we observe empirically.
>
> > The statement that the gap between WGM and the policy baselines is smaller in the missing mass case compared to the $\ell_0$ cardinality case is misleading. While the gap is 5 percentage points (MM is measured as a percentage), WGM performs up to 50% worse than the baselines in some instances when measured as a ratio. This is the apples-to-apples comparison to make when commenting that "for cardinality... sequential methods often output $\approx 2X$ more items [than WGM]."
>
> Taking multiplicative difference in cardinality as a starting point ("sequential methods often output $\approx 2X$ more items [than WGM]"), a more direct alternative comparison may be multiplicative difference in "found" mass, i.e. the mass of the output set, as this more closely corresponds to the cardinality of the output set. If we revisit Figure 1 using found mass, we see that Weighted Gaussian's found mass varies through ~0.8 to ~0.9, and policy methods' found mass varies through ~0.85 to ~0.95. This is, roughly, a multiplicative difference of 0.85 / 0.8 = 1.06 to 0.95 / 0.9 = 1.05, which we suggest is still a genuinely small gap. If desired, we can add a small discussion clarifying this observation.
>
> > Comparison to more baseline algorithms would also be interesting. In the theoretical results, it is very reasonable to only analyze WGM as it is a well-studied algorithm and other candidate algorithms may be very difficult to analyze. On the other hand, if an important empirical takeaway is that "WGM obtains MM within 5% of that of the policy mechanisms, in spite of their significantly more intensive computation," it would be useful to compare to other methods of Swanberg et al. and Chen et al. which offer better cardinality results while still using a similar amount of computation to WGM.
>
> Our proposed empirical takeaway is that Weighted Gaussian obtains mass very close to that of the highest-cardinality (i.e., policy) algorithms. We agree that the algorithms of Swanberg et al. and Chen et al., which are ~2x slower than Weighted Gaussian, likely yield similar or better mass than Weighted Gaussian. However, we suggest that, since the utility gap between Weighted Gaussian and the policy algorithms is already small, their exact performance does not significantly change the overall takeaway.
>
> > Missing mass analysis of any other algorithm, for example the weighted Laplace mechanism to start, would be interesting.
>
> Since the WGM is already near optimal, we opted to not analyze other algorithms. That said, this may be an interesting direction for future work.
>
> > Consider the statement "Note that in Theorem 3.3 the missing mass decays as the total number of items N grows. Moreover, as C decreases or s increases, the upper bound on missing mass decreases." Should the second sentence be conditional on  being sufficiently large compared to  and  (which should be the case for reasonable privacy parameter settings).
>
> You are correct. We have changed this in the new revision.

---

> > ### Author Response · Authors · 2025-11-19
> >
> > > An interesting direction would be to show that WGM is close to optimal on Zipfian data for all $\ell_p$ missing mass for $p > 1$. Is there any chance of showing something like this? Similarly, plotting missing mass as $p$  increases would be interesting, especially if the plots were significantly different for different algorithms.
> >
> > Upper bounds for $\ell_p$ losses can be derived by slightly modifying the arguments used to prove Theorem 3.3 and 3.5. In particular, one first needs to prove the analogous version of Lemma C.2 for the pth missing mass. This can be done by following the exact same steps as in Lines 666-683, but upper bounding the pth-norm MM instead of the $\ell_1$ norm MM in Line 685 onward. Finally, one would need to modify the proof of Theorem 3.3 by first using the Triangle inequality to bound $$MM_p(W, S) \leq MM_p(W, \cup_i \tilde{W}_i) + \left(\sum\_{x \in \tilde{M}}(N(x)/N)^p\right)^{1/p}$$
> >  and then using the pth-norm version of Lemma C.2 (along with Lemmas C.3 and C.4) to bound the right two terms. After doing so, one will get an upper bound on $MM_p$ which decays like $(\epsilon N) ^{\frac{1}{sp} - 1}$. Proving a matching lower bound will be more involved, since lower bounds on $MM_p^p$ do not translate to lower bounds on $MM_p$. We leave this as a direction of future work.
> >
> > > Are both changes of (1) considering missing mass and (2) Zipfian data necessary for giving an absolute utility guarantee? Is there a specific technical challenge to analyzing the cardinality of the set output by WGM on Zipfian data?
> >
> > Unfortunately, cardinality bounds are difficult to derive in full generality but there has been recent work on this topic [1]. That said, it is possible to derive a cardinality bound on the WGM for Zipfian data by relying on standard Gaussian concentration bounds. As a final comment, we want to highlight that our choice to study the MM objective is not because analyzing cardinality is too hard, but because it's a natural and practical metric in its own right. For example, unlike cardinality, the MM allows for one to derive guarantees for top-k selection and hitting-set problems.
> >
> > [1] Dick, Travis, et al. "Private Set Union with Multiple Contributions." The Thirty-ninth Annual Conference on Neural Information Processing Systems.

---

### Official Review · Reviewer_XPcs · 2025-11-08

**Soundness:** 3
**Presentation:** 3
**Contribution:** 3
**Rating:** 8
**Confidence:** 3

**Summary:**

This paper studies several problems in differentially private domain discovery, such as missing mass, top-k selection, and k-hitting set. The authors show that the weighted Gaussian mechanism is useful for these problems. The authors also prove utility lower bounds, showing the algorithm enjoys some level of optimality. The algorithm is tested on several real-world datasets.

**Strengths:**

1. The proposed algorithms based on the weighted Gaussian mechanism are generally applicable to many problems for domain discovery, including finding a set of elements with small missing mass, top-k selection,and k-hitting set.
2. The lower bound results demonstrate that the algorithms are optimal to some extent. For the missing mass problem, the algorithm achieves near-optimal error in terms of privacy parameter $\epsilon$ and total number of items $N$. For top-k selection and k-hitting set, the lower bound is at least $k/\epsilon$, while the upper bounds have an additional $k^{3/2}/\epsilon$ term,
3. Experiments are performed on many real-world datasets, and the performance is comparable or better than previous baseline algorithms.

**Weaknesses:**

1. It seems that the performance of the algorithm depends on the choice of user contribution $\Delta_0$, which ideally has to be close to $\max_|W_i|$. This requires prior knowledge of the maximum elements, or at least some estimate of the parameter of the Zipf distribution of the underlying dataset.
2. There is a mismatch in the upper and lower bounds for top-k selection and k-hitting set.

**Questions:**

1. If a good bound on $\max_|W_i|$ or the $s$ parameter of the Zipf distribution is not known a priori, is there a good way to privately find a good $\Delta_0$ for the WGM mechanism?

---

> ### Author Response · Authors · 2025-11-19
>
> Thanks for the review! We've attempted to address the most salient comments below. Please let us know if further questions arise.
>
> > The performance of the Weighted Gaussian algorithm depends on the choice of $\Delta_0$. This requires prior knowledge of the maximum elements, or at least some estimate of the parameter of the Zipf distribution of the underlying dataset.
>
>  Weighted Gaussian does require choosing $\Delta_0$. Theorem 3.3 shows that the MM grows only logarithmically in $\Delta_0$, hence in practice, one only needs to know its order. Our choice $\Delta_0 = 100$ for most experiments matches the default used in the literature [1, 2]. That said, one can use existing techniques to pick the user contribution bound in a data-dependent fashion. As a concrete example, one can use the unbounded quantile algorithm from [3] to estimate a high quantile (e.g. 0.95) of the user $l_0$ norm distribution, sacrificing a small portion of the privacy budget.
>
> > Mismatch in upper and lower bounds for top-k selection and hitting set
>
> For top-k selection, it is known that the gap of $O(k^{3/2})$ between the true and selected top-k scores is tight, as pointed out by Reviewer sdpD. We will make a note of this in the final version. We leave closing the gap between our upper and lower bounds for hitting set as an interesting direction for future work.
>
> [1] Gopi, Sivakanth, et al. "Differentially private set union." International Conference on Machine Learning. PMLR, 2020.
>
> [2] Swanberg, Marika, Damien Desfontaines, and Samuel Haney. "DP-SIPS: A simpler, more scalable mechanism for differentially private partition selection." arXiv preprint arXiv:2301.01998 (2023).
>
> [3] Durfee, David. "Unbounded differentially private quantile and maximum estimation." Advances in Neural Information Processing Systems 36 (2023): 77691-77712.

---

> > ### Comment · Reviewer_XPcs · 2025-11-25
> >
> > Thanks for your response. My evaluation remains the same.

---

### Author Response · Authors · 2025-11-19
**Revised Manuscript**

A revised manuscript has been uploaded. All changes have been marked in blue.

---

### Meta-Review · Area_Chair_uyQZ · 2026-01-07

**Summary:**

This paper studies differentially private domain discovery under user-level privacy by analyzing the weighted Gaussian mechanism. The paper provides absolute bounds on the utility of the standard WGM algorithm via novel theoretical analysis techniques.

The reviewers appreciate the following strengths of the paper:

- S1. The paper provides absolute bounds on the utility of the standard WGM algorithm via novel theoretical analysis techniques, which is a challenging problem in the literature.

- S2. The paper studies the application of the achieved bounds to problems such as top-k selection and the k-hitting set, where differentially private set union is a subroutine.

- S3. The paper is well written and logically structured.

The authors have successfully addressed most of the reviewers’ concerns during the rebuttal. The ratings were all positive before the rebuttal and are likely to be increased or maintained after the rebuttal.

Overall, the paper has positive evaluations, with most of the raised concerns addressed, and the required revisions are provided in the updated paper and are ready for inclusion in the camera-ready version. Thus, the paper is a strong candidate for acceptance and can be considered for an oral presentation.

**Reviewer Concerns:**

The reviewers also raised the following major concerns:

- W1. The paper lacks discussion of several theoretical details, such as the choice of Delta_0, the mismatched upper and lower bounds for top-k selection and the hitting set, the generality of the Zipfian distribution assumption, and alternative mechanisms such as the weighted Laplace mechanism.

- W2. The paper lacks discussion of the motivation, interpretation, and several experimental details, such as the performance gap compared to baselines.

- W3. The paper could benefit from more experimental results, such as comparisons on synthetic Zipfian data, illustrations of the frequency distribution on real data, reporting the distribution size of |W_i|, and comparisons with more baseline algorithms.

The authors have successfully addressed most of these concerns during the rebuttal period. In particular, the authors have provided additional discussions and clarifications, new experimental results, and an updated manuscript, as detailed below.

- R1. The authors provided further clarifications and the required discussion on the theoretical concerns, addressing W1.

- R2. The authors provided the required discussions on the experimental concerns, addressing W2.

- R3. The authors provided new experimental results, such as plotting the frequency distribution on real datasets, log-log plots of frequency versus rank, and histograms of user item set sizes for each dataset, addressing W3.

The remaining concerns not explicitly addressed are: (1) alternative mechanisms and (2) experiments on new datasets. For (1), the authors emphasize that WGM is already near-optimal and therefore opted not to analyze other algorithms. For (2), the authors provided alternative, less time-consuming experimental results to partially address the concern.

**Reviewer Scores:**

The ratings were all positive before the rebuttal and are likely to be increased or maintained after the rebuttal.

---

### Decision · Program_Chairs · 2026-01-26

Accept (Oral)